# Mean Estimation from Coarse Data: Characterizations and Efficient Algorithms

**Alkis Kalavasis**
Yale University
alkis.kalavasis@yale.edu

**Anay Mehrotra**
Stanford University
anaymehrotra1@gmail.com

**Manolis Zampetakis**
Yale University
manolis.zampetakis@yale.edu

**Felix Zhou**
Yale University
felix.zhou@yale.edu

**Ziyu Zhu**
IMC Trading
zhzy017@gmail.com

## Abstract

Coarse data arise when learners observe only partial information about samples; namely, a set containing the sample rather than its exact value. This occurs naturally through measurement rounding, sensor limitations, and lag in economic systems. We study Gaussian mean estimation from coarse data, where each true sample $x$ is drawn from a $d$-dimensional Gaussian distribution with identity covariance, but is revealed only through the set of a partition containing $x$. When the coarse samples, roughly speaking, have "low" information, the mean cannot be uniquely recovered from observed samples (*i.e.*, the problem is not *identifiable*). Recent work by Fotakis et al. (2021) established that *sample*-efficient mean estimation is possible when the unknown mean is *identifiable* and the partition consists of only *convex* sets. Moreover, they showed that without convexity, mean estimation becomes NP-hard. However, two fundamental questions remained open: (1) When is the mean identifiable under convex partitions? (2) Is *computationally* efficient estimation possible under identifiability and convex partitions? This work resolves both questions. We provide a geometric characterization of when a convex partition is identifiable, showing it depends on whether the convex sets form "slabs" in a direction. Second, we give the first polynomial-time algorithm for finding $\varepsilon$-accurate estimates of the Gaussian mean given coarse samples from an unknown convex partition, matching the optimal $O(d/\varepsilon^2)$ sample complexity. Our results have direct applications to robust machine learning, particularly robustness to observation rounding. As a concrete example, we derive a sample- and computationally- efficient algorithm for linear regression with market friction, a canonical problem in using ML in economics, where exact prices are unobserved and one only sees a range containing the price (Rosett, 1959).

## 1 Introduction

The rapid growth of data-driven inference and decision-making has amplified the need for statistical and algorithmic methods that can robustly handle incomplete or imperfect data. While a large body of statistical literature focuses on the problem of *missing data* (Little & Rubin, 1989; 2019), where observations are either fully precise or totally incomplete, many applications present a different challenge: data are often neither entirely missing nor perfectly observed. For instance, consider healthcare analytics (Chandel et al., 2024; Nietert et al., 2006), recommendation systems (Hu et al., 2008), and sensor networks (Jayasekaramudeli et al., 2024; Dogandžić & Qiu, 2008), where observations are frequently *rounded* to some precision. This occurs when a healthcare form, user feedback system, or sensor network only records values up to the nearest unit (for instance, a rating of $4.7$ is rounded to $5$, only revealing that the true rating lies in $[4.5, 5.5)$). Economics provides another compelling example: market *friction* (Rosett, 1959), where agent responses to environmental changes exhibit lag. Specifically, small movements in underlying conditions fail to trigger price adjustments; instead, prices remain fixed within an "inaction band" (Rosett, 1959). Here, the analyst learns only

that the true, friction-free value lies somewhere within that band, but does not gain any information about the precise value. Similar phenomena arise in physical sciences and survey statistics due to sensor limitations and data aggregation (Wang & Heitjan, 2008; Sheppard, 1897).

These scenarios share a common structure: the statistician observes a subset $S$ of the space containing the true value $v$ but not $v$ itself. The missing data framework cannot capture this partial observability, as it assumes that the data is either fully observed or absent. In contrast, coarse data provides an intermediate case: it restricts $v$ to the observed set $S$, offering some information while still leaving some uncertainty. This fundamental distinction motivated extensive works in statistics (Heitjan & Rubin, 1990; 1991; Heitjan, 1993; Gill et al., 1997) (also see the survey by Amemiya (1973)) to model coarse data and study estimation problems in both supervised and unsupervised settings.

Despite this extensive theoretical foundation, statistically and computationally efficient algorithms for even fundamental tasks have remained elusive in the coarse data setup. In this work, we revisit perhaps the most fundamental problem in this area: *Gaussian mean estimation from coarse data.*

**Problem Setup.** The problem is parameterized by a (potentially unknown) partition $\mathscr{P}$ of $\mathbb{R}^d$ into disjoint sets (Heitjan & Rubin, 1990). For instance, with $d = 1$, $\mathscr{P}$ can be the set of unit intervals formed by the usual grid of $\mathbb{R}$ (namely, $\dots, [-1, 0), [0, 1), [1, 2), \dots$), capturing the coarsening arising from rounding in different aforementioned domains. The learner observes sets $P_1, P_2, P_3, \dots$ from $\mathscr{P}$, where each $P_i$ is sampled as follows:

First sample an (unseen) $x_i \sim \mathcal{N}(\mu^\star, I)$, then output the unique set $P_i$ in $\mathscr{P}$ containing $x_i$.

Our goal is to find an estimate $\widehat{\mu}$ of the unknown mean $\mu^\star$ in time polynomial in both the dimension $d$ and the inverse desired accuracy $1/\varepsilon$.

To gain some intuition, first consider the task without coarsening: given samples $x_1, x_2, \dots, x_n$ from a Gaussian with mean $\mu^\star$, the empirical mean $\sum_i x_i/n$ is $O(1/\sqrt{n})$-close to $\mu^\star$ with high probability. Under coarsening, however, we observe sets instead of points, making this approach inapplicable. A natural idea is to pick a canonical value for each set $P_i$ (such as its centroid) and average these canonical values, but *any* such choice of canonical value can yield estimates $\Omega(r)$ away from $\mu^\star$, where $r$ is the radius of the sets.[1] Thus, Gaussian mean estimation from coarsened data requires more sophisticated methods to achieve vanishing error as the number of samples $n \to \infty$.

Coarse samples make mean estimation strictly more challenging. For some extreme partitions $\mathscr{P}$, estimation becomes impossible. Consider *e.g.*, $\mathscr{P} = \{\mathbb{R}^d\}$, where the partition consists of only the whole space. The learner observes samples $\mathbb{R}^d, \mathbb{R}^d, \dots$, which contain no information about $\mu^\star$, making estimation information-theoretically impossible. A partition is said to be *identifiable* if the mean $\mu^\star$ can be uniquely determined from coarse samples. In the example above, the partition is not identifiable. See Figure 1 for further examples of partitions and their identifiability properties.

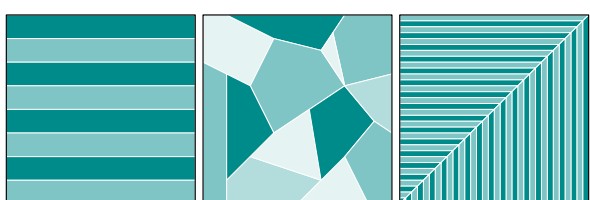

Figure 1: Different partitions $\mathscr{P}$ of $\mathbb{R}^2$. (a) Left: A non-identifiable partition where any $\mu^\star \in \mathbb{R}^2$ and its translation along the $x$-axis induce identical distributions of coarse samples. (b) Middle and (c) right: Identifiable partitions of the space.

**The Need for Convex Partitions.** As mentioned, there are several prior works on learning from coarse data. The key prior work for coarse Gaussian mean estimation is (Fotakis et al., 2021). They showed that the partition structure affects both the identifiability and the computational complexity of mean estimation. In particular, they showed that if $\mathscr{P}$ contains non-convex sets,[2] then Gaussian mean estimation from coarse samples is NP-hard in general. Thus, for computational efficiency, one must focus on *convex partitions*: partitions whose sets are all convex.

---

[1]The radius of a set is the radius of the smallest ball containing the set.

[2]A set $S$ is convex if for any $x_1, x_2 \in S$, the midpoint $(x_1 + x_2)/2$ is also in $S$. Convex sets are quite general and include polytopes (boxes, halfspaces) and ellipsoids.

**Prior Sample Complexity Results.** For convex and identifiable partitions, Fotakis et al. (2021) provided a (non-efficient) method achieving an $\varepsilon$-accurate estimate using $\tilde{O}(d/\varepsilon^2)$ coarse samples.[3] This only establishes sample complexity guarantees, leaving two fundamental questions open:

**Q1.** Which convex partitions result in an identifiable instance?

**Q2.** Is computationally-efficient mean estimation possible for all identifiable convex partitions?

Answering **Q1** contributes to the line of work characterizing identification conditions for missing data problems across statistics (*e.g.*, Everitt & Hand (2013); Teicher (1963)), statistical learning theory (*e.g.*, Angluin & Smith (1983); Cai et al. (2025)), and econometrics (*e.g.*, Manski (1990); Athey & Haile (2002)). **Q2** addresses a key computational question: given Fotakis et al. (2021)'s result on sample-efficient estimation with convex partitions in high dimensions, is there an efficient algorithm for coarse Gaussian mean estimation or do we face a statistical-to-computational gap?

**Our Contributions.** Our main contribution is to completely resolve the above questions:

- ▶ **Identifiability Characterization (Theorem 3.1):** We characterize the convex partitions $\mathscr{P}$ under which the mean $\mu^\star$ is identifiable.
- ▶ **Efficient Algorithm (Theorem 3.2):** We provide the first polynomial-time algorithm for estimating $\mu^\star$ for any convex, identifiable partition $\mathscr{P}$. The algorithm matches the sample complexity of Fotakis et al. (2021), while also being computationally efficient.

These results settle the computational complexity of coarse Gaussian mean estimation under convex partitions. As a concrete application of our techniques, we obtain a sample- and computationally-efficient algorithm for linear regression with market friction, a canonical setting where coarsening arises from lag in market participants' responses (Theorem 3.3).

**Characterization Techniques.** Our first result is a geometric characterization of the sets of the partition $\mathscr{P}$ that allows for identifying the Gaussian mean from coarse observations. Returning to Figure 1(a), it is not difficult to see that partitions that consist of parallel slabs are *not* identifiable. This gives a very natural and intuitive sufficient condition. It remains to argue that this is also necessary, *i.e.*, that *any* non-identifiable partition should look like a collection of parallel slabs.

- ▶ **Necessity of Parallel Slabs:** Our main technical contribution in characterization is to show that parallel slabs are also necessary for non-identifiability. To show this, our proof combines tools from optimal transport and variance reduction inequalities (Vempala, 2010; Hargé, 2004); we believe this combination will have broader applications in estimation with coarse samples.

**Algorithmic Techniques.** Our computationally efficient algorithm performs stochastic gradient descent (SGD) on a variant of the log-likelihood objective (Section 4). Obtaining provable guarantees for this approach requires several key innovations (compared to, *e.g.*, Fotakis et al. (2021)):

- ▶ **Controlling Second Moment:** Establishing SGD convergence requires bounding the second moment of gradients, which intuitively controls the "noise" in each update. In our setting, the second moment scales with the radius of coarse observations and can be unbounded if the partition contains unbounded sets. To overcome this, we introduce an idealized class of *local partitions* (where all sets have a bounded radius; Appendix B.1), develop an algorithm for this class, and then show that a general partition can be reduced to a local one (Appendix B.5).

We refer the reader to Section 4 for further discussion of our techniques and challenges.

## 1.1 RELATED WORKS

In this section, we provide an overview of further related works.

**Learning from Coarse Data.** Inference from coarse data has been extensively studied in statistics (Heitjan & Rubin, 1990; 1991; Heitjan, 1993; Gill et al., 1997; Gill & Grünwald, 2008). The closest to our work is that of Fotakis et al. (2021), who studied coarse estimation in both supervised

---

[3]To be more concrete, the sample complexity scales as $O(d/\varepsilon^2) \cdot \alpha^{-4}$, where $\alpha$ quantifies how "informative" the partition is (Proposition B.5). The exact details are postponed to Section 3.

and unsupervised settings. For coarse Gaussian mean estimation, they give a sample-efficient algorithm for identifiable convex partitions. In the absence of convexity, they show that even fitting the true coarse distribution in total variation distance becomes NP-hard. In PAC learning with multiple labels, they show that any SQ algorithm over "fine" (*i.e.*, non-coarse) labels can be simulated using coarse labels, with coarse sample complexity polynomial in the information distortion and number of fine labels. More recently, Diakonikolas et al. (2025) established an SQ lower bound for linear multiclass classification under coarse labels (under a weaker notion of information preservation). As we have mentioned, coarse observations arise in a number of natural scenarios. An interesting example of this is given by Kalavasis et al. (2025): They showed that the model of Roy (1951) can be framed as a problem of Gaussian mean-estimation with coarse observations arising from a *non-convex* partition. While, in general, non-convex partitions are intractable, the partitions arising in Roy (1951)'s model are highly structured and this enables Kalavasis et al. (2025) to develop an efficient local convergence algorithm for the problem.

**Sheppard's Corrections.**  Sheppard's corrections are approximate corrections to estimates of moments computed from binned data (Sheppard, 1897), *i.e.*, 1-dimensional coarse Gaussian samples from equal-length interval partitions. For mean estimation, our model extends Sheppard's corrections into the high-dimensional setting with non-uniform convex partitions.Our work also relates to Sheppard's corrections, which are classical bias adjustments for *one-dimensional* coarse data (*e.g.*, values rounded to the nearest integer) that yield asymptotically unbiased moment estimators under certain assumptions on the partition (Sheppard, 1897). These corrections are widely used in astronomy and official statistics, among other fields (Schneeweiss & Komlos, 2009; Schneeweiss et al., 2010). For mean estimation, our computationally efficient algorithm generalizes Sheppard's approach to significantly more complex partitions, including high-dimensional ones (Proposition B.5). In addition, we also characterize the statistical limits of estimation with coarse data by characterizing identifiability (Theorem 3.1).

**Learning from Partial Labels.**  Our problem falls in the broader category of learning from partial labels, which has been extensively studied in supervised learning (Jin & Ghahramani, 2002; Durand et al., 2019; Nguyen & Caruana, 2008; Cour et al., 2011; Ratner et al., 2017; De Sa et al., 2016; Cauchois et al., 2024; Feng et al., 2020; Liu & Dietterich, 2014; Feng & An, 2019). Our work studies the unsupervised version of the problem, as we previously discussed.

## 2 PRELIMINARIES

In this section, we present the notation we use and introduce the coarse observational model.

**Notation.** We begin by introducing the notation used throughout the paper.

- ▶ **Vectors and Balls.** We use standard notations for vector norms: For a vector $v \in \mathbb{R}^d$, we use $\|v\|_2$ to denote its $L_2$-norm and $\|v\|_\infty$ its $L_\infty$-norm. We use $B(v, R)$ to denote the $L_2$-ball centered at vector $v$ with radius $R$ and $B_\infty(v, R)$ for the associated $L_\infty$-ball. Also, for $v \in \mathbb{R}^d$ and $i \in [d]$, $v_{-i} \in \mathbb{R}^{d-1}$ denotes the vector obtained from $v$ by removing the $i$-th coordinate.

- ▶ **Partitions.** We usually denote by $\mathscr{P}$ a partition of $\mathbb{R}^d$ and by $P$ an element of this partition. We will call a partition convex if each $P \in \mathscr{P}$ is convex.

- ▶ **Gaussians, Coarse Gaussians, and Distances Between Distributions.** We use $\mathcal{N}(\mu, \Sigma)$ to denote the Gaussian distribution with mean $\mu \in \mathbb{R}^d$ and covariance $\Sigma \in \mathbb{R}^{d \times d}$. In particular, $\mathcal{N}(\mu, \Sigma)$ has density $\mathcal{N}(\mu, \Sigma; x) \propto e^{-\frac{1}{2}(x-\mu)\Sigma^{-1/2}(x-\mu)}$. The mass of some set $S \subseteq \mathbb{R}^d$ under this distribution is denoted by $\mathcal{N}(\mu, \Sigma; S)$. More broadly, for a distribution $p$ and set $S$, $p(S)$ is the mass of $S$ under $p$. For some partition $\mathscr{P}$, we use the notation $\mathcal{N}_\mathscr{P}(\mu, \Sigma)$ to denote the coarse Gaussian distribution, defined as $\mathcal{N}_\mathscr{P}(\mu, \Sigma; P) = \int_P \mathcal{N}(\mu, \Sigma; x) \, dx$ for each $P \in \mathscr{P}$. For two distributions $p, q$ over $\mathbb{R}^d$, we denote their total variation distance as $d_{\mathsf{TV}}(p, q) = (1/2) \int_{\mathbb{R}^d} |p(x) - q(x)| \, dx$ and their KL divergence as $\mathsf{KL}(p\|q) = \mathbb{E}_{x \sim p}[\log(p(x)/q(x))]$.

## 2.1 THE COARSE OBSERVATIONS MODEL

We now formalize the random process that generates coarse observations in Definition 1 and then discuss what it means for a problem instance to be identifiable.

**Definition 1** (Coarse Mean Estimation (Fotakis et al., 2021)). *Let $\mathscr{P}$ be an unknown partition of $\mathbb{R}^d$. Consider the Gaussian distribution $\mathcal{N}(\mu^\star, I)$, with mean $\mu^\star \in \mathbb{R}^d$ (unknown to the learner) and identity covariance matrix. A coarse sample is generated as follows:*

1. *First, nature draws $x$ from $\mathcal{N}(\mu^\star, I)$.*

2. *Then, it reveals the **unique** set $P \in \mathscr{P}$ that contains $x$ (but not $x$ itself) to the learner.*

*We denote the distribution of the coarse sample $P$ as $\mathcal{N}_{\mathscr{P}}(\mu^\star, I)$.*

Given a desired accuracy $\varepsilon > 0$ and coarse samples $P_1, P_2, \ldots$ from the generative process of Definition 1, the learner's goal is to compute an estimate $\widehat{\mu}$ of $\mu^\star$ that, with high-probability, satisfies $\|\widehat{\mu} - \mu^\star\|_2 \leq \varepsilon$ in time that is polynomial in $1/\varepsilon$ and the dimension $d$.

Since the learner's goal is to recover the *parameter* $\mu^\star$, we must at least ensure that the problem is *identifiable*, which roughly means that parameter recovery is possible for any $\varepsilon > 0$. Recall that there are some extreme partitions, such as $\mathscr{P} = \{\mathbb{R}^d\}$, for which this is impossible. The following notion of information preservation is equivalent to identifiability (and we explain this after the definition).

**Definition 2** (Information Preservation (a.k.a. Identifiability)). *A partition $\mathscr{P}$ is said to be information preserving with respect to $\mu^\star$ if for any $\mu \neq \mu^\star$, it holds that $d_{\mathsf{TV}}(\mathcal{N}_{\mathscr{P}}(\mu^\star, I), \mathcal{N}_{\mathscr{P}}(\mu, I)) > 0$.*

Recall that the total variation or TV distance between two distributions is 0 if and only if the two distributions are identical. Thus if $\mathscr{P}$ is not information-preserving, there are two distinct mean values $\mu^\star \neq \mu$ such that the corresponding coarse distributions $\mathcal{N}_{\mathscr{P}}(\mu^\star, I)$ and $\mathcal{N}_{\mathscr{P}}(\mu, I)$ are identical, making the two indistinguishable even with infinite coarse samples. Conversely, when a partition is information-preserving, then the coarse distributions induced by any $\mu \neq \mu^\star$ differ in total variation. Thus infinitely many coarse samples suffice to distinguish $\mu^\star$ from any other $\mu \neq \mu^\star$.

As examples, we have already seen that some partitions are not identifiable; examples include the partition $\mathscr{P} = \{\mathbb{R}^d\}$ and the partition in Figure 1(a); while those in Figure 1(b,c) are identifiable.

Given the identifiability of an instance, one can study the number of samples and time needed to estimate the mean. For that, one needs to introduce a quantitative version of the identification condition and, indeed, to this end, prior work has introduced the notion of $\alpha$-*information preserving*:

**Definition 3** ($\alpha$-Information Preservation; (Fotakis et al., 2021)). *Given $\alpha \in (0, 1]$, we say that the partition $\mathscr{P}$ of $\mathbb{R}^d$ is $\alpha$-information preserving with respect to $\mu^\star$ if for any $\mu \neq \mu^\star$, it holds that*

$$d_{\mathsf{TV}}(\mathcal{N}_{\mathscr{P}}(\mu^\star, I), \mathcal{N}_{\mathscr{P}}(\mu, I)) \geq \alpha \cdot d_{\mathsf{TV}}(\mathcal{N}(\mu^\star, I), \mathcal{N}(\mu, I)) . \tag{1}$$

As a sanity check, note that for any $\mu \neq \mu^\star$, we have $d_{\mathsf{TV}}(\mathcal{N}(\mu^\star, I), \mathcal{N}(\mu, I)) > 0$, so $\alpha$-information-preservation for any $\alpha > 0$ implies information preservation. Fotakis et al. (2021)'s sample-efficient algorithms require this stronger $\alpha$-information-preservation notion. Our computationally efficient algorithm (Theorem 3.2) will also require this strengthening of identifiability.

## 3 OUR RESULTS

In this section, we present our main results. Section 3.1 presents the characterization of identifiability, Section 3.2 the computationally-efficient algorithm, and, finally, Section 3.3 presents the application of the computationally-efficient algorithm to regression with friction.

## 3.1 CHARACTERIZATION OF IDENTIFIABILITY FOR CONVEX PARTITIONS

In this section, we give a geometric characterization of when a convex partition is identifiable. Informally, a partition fails to be identifiable if and only if it is translation invariant in some direction. To formally state the characterization (and prove it), we need the following notion of a "slab."

**Definition 4** (Slab). *Given unit vector $v \in \mathbb{R}^d$, a (non-empty) set $P \subseteq \mathbb{R}^d$ is said to be a* slab *in direction $v$ if $P$ is invariant under translations along direction $v$. That is, $P$ is a slab in direction $v$ if it has the property that if $x \in P$ then $x + tv \in P$ for all $t \in \mathbb{R}$.*

Examples of slabs include all the sets in the partition in Figure 1(a) and the sets in Figure 2. Note that because of the translation invariance, any slab in direction $v$ must be unbounded in that direction. Hence, bounded sets (such as the unit circle or a bounded box) are not slabs in any direction.

Now we are ready to formally state our characterization. Its proof appears in Appendix A.

**Theorem 3.1** (Characterization of identifiability for convex partitions). *Fix any $\mu^\star \in \mathbb{R}^d$. A convex partition $\mathscr{P}$ of $\mathbb{R}^d$ is **not** information preserving (i.e., **not** identifiable; Definition 2) with respect to $\mu^\star$ if and only if there is a unit vector $v \in \mathbb{R}^d$ such that almost every[4] $P \in \mathscr{P}$ is a slab in direction $v$.*

We have already seen one example of a non-identifiable partition in Figure 1(a) in two dimensions. One can also construct examples in higher dimensions, *e.g.*, partitions in three dimensions using the slabs in Figure 2(a). Some more remarks are in order.

1. Information preservation (*i.e.*, identifiability) does not depend on $\mu^\star$: either $\mathscr{P}$ is identifiable for every $\mu^\star$ or for none. Hence, the difficulty of coarse Gaussian mean estimation comes from the structure of the coarse samples and not the specific Gaussian distribution.

2. The characterization in Theorem 3.1 is already sufficient to deduce identifiability with respect to several interesting partitions. For instance, consider the partition induced by rounding, *i.e.*, the grid partition $\mathscr{P}$ of $\mathbb{R}^d$ formed by axis-aligned boxes of side length, say, 1. This is identifiable by Theorem 3.1 because every $P \in \mathscr{P}$ is convex and bounded (not slabs).

3. The "almost every" quantifier in Theorem 3.1 is important: some convex partitions $\mathscr{P}$ contain non-slabs but are non-identifiable because the non-slabs have zero total volume. For instance, consider the following partition of $\mathbb{R}^2$ by (infinite) vertical lines $\ell_a := \{(a, y) : y \in \mathbb{R}\}$ at each $x = a$ for $a \neq 1$, and at $x = 1$ by two half-lines $\ell_{1,+} := \{(1, y) : y \geq 0\}$ and $\ell_{1,-} := \{(1, y) : y < 0\}$. Here $\ell_{1,+}$ and $\ell_{1,-}$ are non-slabs, yet the partition is non-identifiable because these sets together have zero volume. Intuitively, manipulating zero-volume sets does not change the coarse distribution in total variation distance (see Definition 1).

4. Finally, we note that identifiability is purely information-theoretic. Being identifiable does not by itself guarantee sample-efficient estimation. For this, the natural strengthening of information preservation, $\alpha$-information preservation, is required (Definition 3).

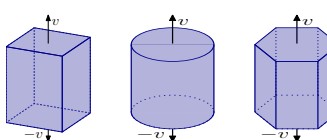

Figure 2: An illustration of different slabs in direction $v$ in three dimensions. Intuitively, a slab in direction $v$ is formed by taking a 2D shape ((a) square, (b) circle, (c) hexagon in each of the three subfigures) and extending it infinitely in direction $v$ (and $-v$).

## 3.2 Polynomial-Time Gaussian Mean Estimation from Convex Partitions

Next, we present our computationally efficient algorithm showing that polynomial-time coarse Gaussian mean estimation is possible from convex partitions. The running time of these algorithms degrades naturally with how much information the partition preserves, as captured by the information preservation parameter $\alpha$ (Definition 3).

To be precise about running time, we need to specify how the algorithm accesses a coarse sample $P$. In high dimensions, the representation of the set can dramatically affect the computational complexity of a problem, a well-known issue in geometric algorithms (Grötschel et al., 1988). Our algorithm works with natural encodings (*e.g.*, polytopes described explicitly by linear inequalities, or convex bodies implicitly described by a separation oracle and an inner radius) and runs in polynomial time relative to the *bit complexity* of the encodings of observed sets.[5]

Our computationally-efficient result is as follows. Its proof appears in Appendix B.

---

[4]Let $\mathscr{B} \subseteq \mathscr{P}$ be the set of non-slabs in $\mathscr{P}$. "Almost every" means that the volume of $\bigcup_{P \in \mathscr{B}} P$ is 0.

[5]Bit complexity measures how many bits are needed to encode the input using the standard binary encoding (which, *e.g.*, maps integers to their binary representation, rational numbers as a pair of integers, and vectors/matrices as a tuple of their entries) (Grötschel et al., 1988, Section 1.3)

**Theorem 3.2** (Coarse Mean Estimation Algorithm). *There is an algorithm that, for any true mean $\mu^\star \in \mathbb{R}^d$ with $\|\mu^\star\|_2 \leq D$ and any partition $\mathscr{P}$ that is convex and $\alpha$-information preserving with respect to $\mu^\star$ (Definition 3), gets as input the desired accuracy and confidence $\varepsilon, \delta \in (0, 1)$, draws i.i.d. coarse Gaussian samples from $\mathscr{P}$, and outputs an estimate $\widehat{\mu}$ satisfying*

$$\|\widehat{\mu} - \mu^\star\|_2 \leq \varepsilon$$

*with probability $1 - \delta$. The algorithms uses*

$$m = \widetilde{O}\left(\frac{dD^2 \log 1/\delta}{\alpha^4} + \frac{d \log 1/\delta}{\alpha^4 \varepsilon^2}\right)$$

*coarse samples and runs in time polynomial in $m$ and the bit complexity of the coarse samples.*

Thus, when the information preservation is constant, *i.e.*, $\alpha = \Omega(1)$, and we are provided a constant warm-start to the mean (*i.e.*, $D = O(1)$), the above algorithm obtains an $\varepsilon$-accurate estimate to $\mu^\star$ in $\widetilde{O}(d/\varepsilon^2)$ time while using $\widetilde{O}(d/\varepsilon^2)$ samples, matching the sample complexity of Fotakis et al. (2021) while being computationally efficient. Moreover, we can actually prove a stronger guarantee: we can replace $\|\mu^\star\|_2 \leq D$ in the theorem by the strictly weaker requirement that $\|\mu^\star\|_\infty \leq D$ (this is strictly weaker since $\|\mu^\star\|_\infty \leq \|\mu^\star\|_2$).

**Extension to Mixtures of Partitions Model.**  Some works on coarse estimation consider an extension of the model in Definition 1 where instead of a single partition $\mathscr{P}$, each coarse sample is drawn from a different partition $\mathscr{P}$. Formally, there are convex partitions $\mathscr{P}_1, \ldots, \mathscr{P}_\ell$ and the observations are sets $P$ where $P \in \mathscr{P}_j$ for some $1 \leq j \leq \ell$, where $j$ is chosen independently by some prior distribution over $[\ell]$. The underlying data-generating distribution $\mathcal{N}(\mu^\star, I)$ remains the same but the observations are created by first drawing a partition index $1 \leq j \leq \ell$ according to some mixture probability, independently drawing $x \sim \mathcal{N}(\mu^\star, I)$, and then outputting $P \in \mathscr{P}_j$ containing $x$. We remark that our algorithm, like the sample-efficient method of Fotakis et al. (2021), also straightforwardly extends to this model with the same time and sample complexity.

### 3.3 Application to Regression with Friction

Next, to illustrate the flexibility of our algorithmic techniques, we provide an application to linear regression with market friction, a problem dating back to Rosett (1959). We present our main result and an informal description of the model here. The formal statement of the result, its proof, and a formal description of the model appear in Appendix E.

**Setup: Linear Regression with Market Friction.**  Consider the task of analyzing the relationship between changes in the yield of a bond $x \in \mathbb{R}$ and the change in an investor's bond holdings $y \in \mathbb{R}$. In an ideal world, we would expect a linear relationship between changes in yield and changes in holdings $y = x \cdot w^\star + \xi$ for $\xi \sim \mathcal{N}(0, 1)$. However, due to transaction costs, bid-ask spreads, and the time and effort required to execute a trade, investors will not adjust their holdings for small changes in the bond's yield. Thus, even if the underlying relationship is linear, we only observe a transformation $(x, z = c(y))$ by some friction function $c : \mathbb{R} \to \mathbb{R}$ (Figure 3(a)).

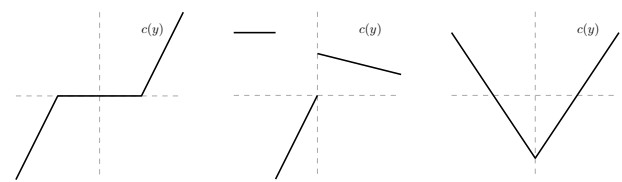

Figure 3: Illustrations of different friction functions $c : \mathbb{R} \to \mathbb{R}$. (a) Left: monotone with convex pre-images. (b) Middle: not monotone but has convex pre-images. (c) Right: not monotone and has non-convex pre-images.

**Translating Market Friction to Coarsening.**  To utilize our algorithmic techniques, we need to frame the problem as an instance of coarsening. Indeed, we can think of an observation pair $(x, z = c(y))$ as the coarse observation $(x, S)$ where $S = c^{-1}(z) \subseteq \mathbb{R}$. This gives us a bridge to apply the techniques we developed for coarse mean estimation. As in the case of coarse mean estimation, it may be information theoretically impossible to estimate $w^\star \in \mathbb{R}^d$ for all friction functions. Thus,

we require an analogous notion of information preservation for instances of linear regression with friction (Assumption 4) and design an algorithm for estimating $w^\star$ when the friction function $c$ is information preserving and also has convex pre-images, *i.e.*, $c^{-1}(z) \subseteq \mathbb{R}$ is convex for all $z \in \mathbb{R}$. This is a large class and includes, *e.g.*, monotonically non-decreasing functions as a special case.

**Theorem 3.3** (Informal; See Theorem E.1). *There is an algorithm for $\alpha$-information preserving instances of linear regression with friction that, given $n$ observations, outputs an estimate $\widehat{w} \in \mathbb{R}^d$ satisfying $\|\widehat{w} - w^\star\|_2^2 \leq \widetilde{O}\left(\frac{d}{\alpha^4 n}\right)$ with probability $0.99$ in $\mathrm{poly}(n, d)$ time.*

# 4    OVERVIEW OF MAIN PROOFS

Next, we overview the proofs of our main results and defer the exact details to Appendices A and B.

A central component of both the characterization and the algorithmic result is the log-likelihood function with respect to coarse data. We refer the reader to Rossi (2018); Cover (1999) for useful preliminaries on the log-likelihood. For a partition $\mathscr{P}$ and true mean $\mu^\star$, the negative log-likelihood function arising from the coarse-observations model (Definition 1) is as follows

$$\mathscr{L}(\mu) \;:=\; \mathbb{E}_{P \sim \mathcal{N}_{\mathscr{P}}(\mu^\star, I)} \left[ -\log \mathcal{N}_{\mathscr{P}}(\mu, I; P) \right] .$$

To illustrate the usefulness of the log-likelihood, we present a short proof connecting a certain property of the log-likelihood to identifiability; this is the starting point of our characterization.

**Proposition 4.1** (Likelihood-Based Characterization of Identifiability). *A partition $\mathscr{P}$ of $\mathbb{R}^d$ is information preserving (Definition 2) with respect to $\mu^\star \in \mathbb{R}^d$ if and only if $\mathscr{L}(\cdot)$ has a* unique *minimizer.*

Thus, if $\mathscr{P}$ is identifiable, then the corresponding negative log-likelihood has a unique minimizer $\mathscr{L}(\cdot)$.[6] Further, one can show that $\mu^\star$ is always a minimizer of $\mathscr{L}(\cdot)$. Therefore, Proposition 4.1 also tells us that if a partition is identifiable, then the unique minimizer of $\mathscr{L}(\cdot)$ is $\mu^\star$ itself.

*Proof of Proposition 4.1.* Suppose the instance is identifiable. The following holds for any $\mu \neq \mu^\star$,

$$\mathscr{L}(\mu) - \mathscr{L}(\mu^\star) = \mathsf{KL}(\mathcal{N}_{\mathscr{P}}(\mu^\star, I) \,\|\, \mathcal{N}_{\mathscr{P}}(\mu, I)) \geq 2\, \mathrm{d}_{\mathsf{TV}}(\mathcal{N}_{\mathscr{P}}(\mu^\star, I), \mathcal{N}_{\mathscr{P}}(\mu, I))^2 > 0 .$$

Here, the first equality is a standard property of the log-likelihood (Cover, 1999), the second is Pinsker's inequality, and the third is the definition of identifiability of the instance.

On the other hand, if the instance is not identifiable, we can find some $\mu \neq \mu^\star$ for which $\mathrm{d}_{\mathsf{TV}}(\mathcal{N}_{\mathscr{P}}(\mu^\star, I), \mathcal{N}_{\mathscr{P}}(\mu, I)) = 0$. Therefore, $\mathcal{N}_{\mathscr{P}}(\mu^\star, I) = \mathcal{N}_{\mathscr{P}}(\mu, I)$, which implies that $\mathscr{L}(\mu) - \mathscr{L}(\mu^\star) = 0$ because $\mathsf{KL}(\mathcal{N}_{\mathscr{P}}(\mu^\star, I) \,\|\, \mathcal{N}_{\mathscr{P}}(\mu, I)) = 0$. Hence, $\mu \neq \mu^\star$ is also minimizes $\mathscr{L}(\cdot)$. $\quad\square$

Another useful property of $\mathscr{L}(\cdot)$ is that it is convex whenever the partition $\mathscr{P}$ is convex. This result is due to Fotakis et al. (2021), who proved it using a variance reduction inequality of Hargé (2004).

**Proposition 4.2** (Convexity; (Fotakis et al., 2021)). *If $\mathscr{P}$ is a convex partition, then $\mathscr{L}(\cdot)$ is convex.*

Propositions 4.1 and 4.2 suggest a natural approach for both results: For the characterization, we will characterize partitions for which $\mathscr{L}(\cdot)$ has a unique minimizer. For the algorithmic result, we will develop a computationally efficient algorithm to find $\mathscr{L}(\cdot)$'s minimizer using its convexity (Proposition 4.2). Next, we explain how we implement this approach and the corresponding challenges.

## 4.1    CHARACTERIZATION OF IDENTIFIABILITY (THEOREM 3.1)

Now we are ready to give an overview of the proof of Theorem 3.1. This has two directions. The harder direction is to show that when the coarse mean estimation problem is not identifiable under a convex partition $\mathscr{P}$, then the sets of $\mathscr{P}$ must be parallel slabs.

Since the instance is not identifiable, Proposition 4.1 implies that $\mathscr{L}(\cdot)$ has two distinct minimizers. As we saw above, $\mu^\star$ is always a minimizer. Now, using the convexity of $\mathscr{L}(\cdot)$ (Proposition 4.2),

---

[6]This follows because of the classic equality $\mathscr{L}(\mu) = \mathsf{KL}(\mathcal{N}_{\mathscr{P}}(\mu^\star, I) \,\|\, \mathcal{N}_{\mathscr{P}}(\mu, I)) + \mathrm{const}$ (Cover, 1999). Indeed, since the KL is always non-negative and it achieves a value 0 at $\mu^\star$, $\mu^\star$ must be a minimizer of $\mathscr{L}(\cdot)$.

it can be shown that: there exists a unit vector $u$ for which $u^\top \boldsymbol{\nabla}^2 \mathscr{L}(\mu^\star)\, u \;=\; 0$. Substituting the formula for the Hessian (derived in Appendix B) into the above implies:

$$1 \;=\; \mathop{\mathbb{E}}_{P \sim \mathcal{N}_{\mathscr{P}}(\mu^\star, I)} \left[ \mathop{\mathrm{Var}}_{x \sim \mathcal{N}(\mu^\star, I)\,|\,x \in P} \left[ u^\top x \right] \right].$$

We divide the rest of the overview into two parts.

**Step I (Variance Reduction Inequality).** From the variance reduction properties of Gaussian distributions (see *e.g.*, Hargé (2004)), we know in addition that $\mathrm{Var}_{x \sim \mathcal{N}(\mu^\star, I)\,|\,x \in P}[u^\top x] \leq 1$. But since the expectation $\mathrm{Var}_{x \sim \mathcal{N}(\mu^\star, I)\,|\,x \in P}[u^\top x]$ over the draw of $P$ is 1, it must be the case that almost every set $P \in \mathscr{P}$ satisfies

$$\mathop{\mathrm{Var}}_{x \sim \mathcal{N}(\mu^\star, I)\,|\,x \in P} [\langle u, x \rangle] \;=\; 1.$$

**Step II (Combination of Variance Reduction and Prékopa–Leindler Inequalities).** We now make use of the above property to show that almost all sets $P$ of the partition must be slabs in the same direction. To do this, we consider the projection of the points to $u$ and to $u^\perp$. Let $y = u^\top x$ with $x \sim \mathcal{N}(\mu^\star, I)\,(P)$. First, using a variance reduction inequality (Vempala, 2010), we show that $y$ has a Gaussian distribution despite truncation (Lemma A.5). Then, using the Prékopa–Leindler Inequality, we show that this is only possible if $P$ is a slab in the direction of $u$; in particular, if $P$ is of the form $\{t \cdot u \colon t \in \mathbb{R}\} \times C_P$ (Proposition A.6).

We believe that the combination of variance reduction and Prékopa–Leindler-type inequalities can have further applications in characterizing other coarse estimation problems.

## 4.2 Main Algorithmic Ideas (Theorem 3.2)

Given an identifiable convex partition, our goal is to design an efficient algorithm to find $\widehat{\mu} \approx \mu^\star$. From Proposition 4.1 and Theorem D.1, we know that the log-likelihood is convex with unique minimizer $\mu^\star$. Hence, a natural approach is to use stochastic gradient descent (SGD) to find the minimizer. There are, however, several challenges in implementing this, as we discuss below. Before we proceed, we note that while Fotakis et al. (2021) also recognized that the log-likelihood is convex and has a unique minimizer, they used brute force search over an $\exp(d/\varepsilon)$-sized $\varepsilon$-net (see Har-Peled (2011)) of the domain, yielding a sample-efficient but computationally inefficient algorithm.

**Challenge and Idea I (Local Strong Convexity).** Under some appropriate conditions on the second moment of gradients, SGD can find parameter estimates $\widehat{\mu}$ with $\mathscr{L}(\widehat{\mu}) \leq \mathscr{L}(\mu^\star) + \varepsilon$. However, $\widehat{\mu}$ can still be far from $\mu^\star$. If $\mathscr{L}(\cdot)$ was *strongly* convex, then $\widehat{\mu}$ would be close to $\mu^\star$, but, unfortunately, in this setting $\mathscr{L}(\cdot)$ is *not* strongly convex everywhere. Hence, we need to exploit some additional structure of $\mathscr{L}(\cdot)$ to translate the optimality in function value to closeness in parameter distance. Toward this, we show that the log-likelihood is *locally* strongly convex around the true solution $\mu^\star$; this means that the function is strongly convex in a small ball around $\mu^\star$.[7] To prove this, we make use of a natural connection between information preservation and the log-likelihood objective through the KL divergence, together with Pinsker's inequality. These yield the following (Lemma B.3):

$$\mathscr{L}(\mu) - \mathscr{L}(\mu^\star) = \mathop{\mathbb{E}}_{P \sim \mathcal{N}_{\mathscr{P}}(\mu^\star)} \left[ \log \tfrac{\mathcal{N}(\mu^\star; P)}{\mathcal{N}(\mu; P)} \right] = \mathsf{KL}(\mathcal{N}_{\mathscr{P}}(\mu^\star) \| \mathcal{N}_{\mathscr{P}}(\mu)) \geq 2\, \mathrm{d}_{\mathsf{TV}}(\mathcal{N}_{\mathscr{P}}(\mu), \mathcal{N}_{\mathscr{P}}(\mu^\star))^2.$$

Hence, using information preservation (Definition 3) and lower bounds on the total variation distance between two Gaussians (Arbas et al., 2023) implies that (see Lemma B.3)

$$\mathscr{L}(\mu) - \mathscr{L}(\mu^\star) \geq 2\alpha^2\, \mathrm{d}_{\mathsf{TV}}(\mathcal{N}(\mu), \mathcal{N}(\mu^\star))^2 \geq \min\left\{ \Omega(\alpha^2), \Omega(\alpha^2 \, \|\mu - \mu^\star\|_2^2) \right\}.$$

The above ensures that $\mathscr{L}(\cdot)$ is strongly convex around a small region of the true parameter $\mu^\star$.

Having ensured (i) that approximate minimizers in function value are also good estimators, we also need to (ii) bound the second moment of the gradient norm to implement SGD efficiently. To argue about (ii), consider the gradient of $\mathscr{L}(\cdot)$ (see Appendix B.6):

$$\boldsymbol{\nabla}\mathscr{L}(\mu) \;=\; \mu - \mathop{\mathbb{E}}_{P \sim \mathcal{N}_{\mathscr{P}}(\mu^\star, I)} \mathop{\mathbb{E}}_{\mathcal{N}(\mu, I, P)} [x]. \tag{2}$$

---

[7]Note that since we do not know $\mu^\star$ and do not have a starting point within this small ball, we cannot simply run projected stochastic gradient descent.

**Challenge and Idea II (Bounding the Second Moment of the Gradients).** Bounding the second moment of the stochastic gradients is not straightforward since the inner and outer expectations in Equation (2) are not over the same means; in general, these two means can be as far as the diameter of $P$. Because of this, the norm of the gradient can scale with $P$'s, and we must somehow control the gradient norm when the diameter gets large. The key idea here is to use the strong concentration properties of the Gaussian measure in high dimensions. In particular, with extremely high probability, a collection of $m$ i.i.d. draws of the standard normal distribution in $d$ dimensions will fall inside a $L_\infty$-box of radius $O(1)$. This means that if we observe a set $P$ from the unknown coarse Gaussian, while we are agnostic to the specific Gaussian sample, we can be extremely confident that it belongs to $P \cap B_\infty(0, R)$ for some appropriate radius $R$. Now, since we will eventually use $m$ samples, we can take $R = D + O(\log(md/\delta))$ and make the confidence of the learner $1 - \delta$. In more detail, conditioning on these high-probability events on the $m$ samples, we can replace each observed set $P$ with its intersection with the ball $B_\infty(0, R)$, and introduce a new partition of the space (see Appendix B.1). For this new partition, we can obtain upper bounds on the moments of the stochastic gradients and derive an algorithm that recovers the mean. Finally, we show that we can reduce any convex partition to this special case (see Appendix B.4).

## 5 CONCLUSION

In this work, we study the problem of Gaussian mean estimation from coarse samples. Coarse samples arise in many domains (from healthcare analytics, recommendation systems, physical sciences, to economics) and due to many reasons (from rounding, sensor limitations, to response-lag). We focus on the setting where all the coarse samples are convex, as otherwise, the problem is NP-hard (Fotakis et al., 2021). In this setting, we completely settle the computational complexity of the problem by characterizing all convex partitions that are identifiable (Theorem 3.1) and giving a polynomial-time algorithm for identifiable instances (Theorem 3.2). Both of our results use several new techniques that we believe would be of broader interest to other estimation problems from coarse samples (see Section 4). As a concrete application of our algorithmic ideas, we show that our computationally efficient algorithm can be adapted to give an efficient algorithm for linear regression with market friction (Theorem 3.3), a problem dating back to Rosett (1959).

Our work leaves various open questions for future work: While we study Gaussian estimation with a known covariance, understanding the computational complexity of estimation with unknown covariance is an interesting direction. Developing efficient algorithms for this problem would require new ideas since the log-likelihood for this case can be non-convex. Another interesting question is to extend our guarantees beyond Gaussians to other distribution families (see Appendix F for preliminary results in this direction).

## LARGE LANGUAGE MODEL USE

After writing a preliminary version of the paper, we used LLMs to help streamline the writing and find typographical errors.

## ACKNOWLEDGMENTS

Alkis Kalavasis was supported by the Institute for Foundations of Data Science at Yale. Felix Zhou acknowledges the support of the Natural Sciences and Engineering Research Council of Canada (NSERC).

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

CONTENTS

# A    PROOF OF THEOREM 3.1 (CHARACTERIZATION OF IDENTIFIABILITY FOR CONVEX PARTITIONS)

In this section, we prove Theorem 3.1. We will begin with a restatement of the result.

**Theorem 3.1** (Characterization of identifiability for convex partitions). *Fix any $\mu^\star \in \mathbb{R}^d$. A convex partition $\mathcal{P}$ of $\mathbb{R}^d$ is **not** information preserving (i.e., **not** identifiable; Definition 2) with respect to $\mu^\star$ if and only if there is a unit vector $v \in \mathbb{R}^d$ such that almost every[8] $P \in \mathcal{P}$ is a slab in direction $v$.*

Before outlining the proof, we remark that an equivalent definition of being a slab in direction $v$ that is more convenient to work with is that there is a rotation (orthogonal linear map) $R : \mathbb{R}^d \to \mathbb{R}^d$ with $Rv = e_1$ such that
$$RP \ = \ \mathbb{R} \times S \qquad \text{for some } S \subseteq \mathbb{R}^{d-1} \,.$$

**Outline.**    We prove Theorem 3.1 by considering both directions of the implication. The first direction shows that if the mean is non-identifiable, then, outside a null sub-family of sets, each $P \in \mathcal{P}$ must be a slab. This can be broken down into two steps.

1. **Unit variance on partition $\mathcal{P}$.** First, we show that non-identifiability forces the population negative log-likelihood to have a "flat direction" – along which the value of the log-likelihood does not change. Next, we show that along this direction, say $u$, the variance of the one-dimensional projection $\langle u, x \rangle$ under the truncated Gaussian $\mathcal{N}(\mu^\star, I, P)$ equals 1 for almost every $P \in \mathcal{P}$ (i.e., the union of sets $P \in \mathcal{P}$ violating this requirement has measure 0).

2. **From unit variance to slabs (a.e.).** Second, we show that the unit-variance condition implies that almost every $P \in \mathcal{P}$ is a direct sum of the form $\mathbb{R}u \oplus C_P$, i.e., a slab. This step uses the equality case of the Prékopa–Leindler inequality established by Dubuc (1977).

The other direction shows the converse (and easy) direction: if almost every set is a slab along some $v$, then the coarse distribution is invariant under shifts by $tv$, so identifiability fails.

## A.1    FLATNESS ⇒ UNIT-VARIANCE PROJECTIONS

First, we observe that non-identifiability implies that the population negative log-likelihood is "flat" in some direction $u$.

**Lemma A.1** (Zero directional Hessian). *Let $\mathscr{L}(\mu) := \mathbb{E}_{P \sim \mathcal{N}_{\mathcal{P}}(\mu^\star, I)} [-\log \mathcal{N}_{\mathcal{P}}(\mu, I; P)]$. Suppose there exists $\mu_0 \neq \mu^\star$ with $\mathcal{N}_{\mathcal{P}}(\mu_0, I) \equiv \mathcal{N}_{\mathcal{P}}(\mu^\star, I)$. Then for $u := \frac{\mu_0 - \mu^\star}{\|\mu_0 - \mu^\star\|}$,*
$$u^\top \boldsymbol{\nabla}^2 \mathscr{L}(\mu^\star) \, u \ = \ 0 \,.$$

*Proof.* Since $\mathcal{P}$ is a convex partition, $\mathscr{L}$ is also convex (Fotakis et al., 2021). Hence, the restriction $g(t) := \mathscr{L}(\mu^\star + t(\mu_0 - \mu^\star))$ is convex. Since $g(0) = g(1)$, $g$ is constant over $[0, 1]$. In particular, $g''(0) = 0$, which is exactly $u^\top \boldsymbol{\nabla}^2 \mathscr{L}(\mu^\star) u = 0$. $\qquad\square$

Next, we recall that the Hessian of the negative log-likelihood has the following expression (Fotakis et al., 2021)
$$\boldsymbol{\nabla}^2 \mathscr{L}(\mu) \ = \ I \ - \ \mathbb{E}_{P \sim \mathcal{N}_{\mathcal{P}}(\mu^\star, I)} \left[ \operatorname*{Cov}_{x \sim \mathcal{N}(\mu, I, P)}[x] \right] \,. \tag{3}$$

Combining (3) at $\mu = \mu^\star$ with Lemma A.1 yields
$$1 \ = \ \mathbb{E}_{P \sim \mathcal{N}_{\mathcal{P}}(\mu^\star, I)} \operatorname*{Var}_{x \sim \mathcal{N}(\mu^\star, I, P)} [\langle u, x \rangle] \,. \tag{4}$$

For convex $P$, the one-dimensional variance reduction (*e.g.*, via Brascamp–Lieb in 1D, see Lemma A.4) gives
$$\operatorname*{Var}_{x \sim \mathcal{N}(\mu^\star, I, P)} [\langle u, x \rangle] \ \leq \ 1 \,. \tag{5}$$

---

[8]Let $\mathcal{B} \subseteq \mathcal{P}$ be the set of non-slabs in $\mathcal{P}$. "Almost every" means that the volume of $\bigcup_{P \in \mathcal{B}} P$ is 0.

Since (4) is the expectation of a $[0, 1]$-valued random variable, we conclude that equality in (5) holds for $\mathcal{N}_{\mathcal{P}}\left(\mu^{\star}, I\right)$-*almost every* set $P$. Equivalently, the sub-collection of sets with strictly smaller variance has mass 0. We record this in the following lemma.

**Lemma A.2** (Unit variance for almost all sets). *Suppose there exists $\mu_0 \neq \mu^{\star}$ with $\mathcal{N}_{\mathcal{P}}(\mu_0, I) = \mathcal{N}_{\mathcal{P}}\left(\mu^{\star}, I\right)$. Then there exists $u \in \mathcal{S}^{d-1}$ such that*

$$\operatorname*{Var}_{x \sim \mathcal{N}(\mu^{\star}, I, P)}[\langle u, x\rangle] = 1 \quad \text{for almost every } P \in \mathcal{P}.$$

*Equivalently,*

$$\mathcal{N}_{\mathcal{P}}\left(\mu^{\star}, I\right)(S) = 0 \quad \text{where} \quad S = \left\{ P \in \mathcal{P} : \operatorname*{Var}_{x \sim \mathcal{N}(\mu^{\star}, I, P)}[\langle u, x\rangle] < 1 \right\}.$$

### A.2 FROM UNIT VARIANCE TO SLABS (ALMOST EVERYWHERE)

Next, we use the observation from the previous step to show that almost all $P \in \mathcal{P}$ must be slabs.

Fix the unit vector $u$ furnished by Lemma A.2. Rotate coordinates so that $u = e_1$ and write $x = (y, z) \in \mathbb{R} \times u^{\perp}$, $\mu^{\star} = (\mu_1^{\star}, \mu_{\perp}^{\star})$. For a set $P \in \mathcal{P}$, define the set $C_P$ and its Gaussian weight $W_P$

$$C_P(y) := \{ z \in u^{\perp} : (y, z) \in P \}, \qquad W_P(y) := \int_{u^{\perp}} \mathbb{1}_{C_P(y)}(z)\,\varphi_{d-1}(z - \mu_{\perp}^{\star})\,\mathrm{d}z,$$

where $\varphi_{d-1}(w) \propto e^{-\frac{1}{2}\|w\|^2}$ is the $(d-1)$-dimensional standard Gaussian density. The law of $y = \langle u, x\rangle$ under $x \sim \mathcal{N}(\mu^{\star}, I, P)$ has density

$$f_P(y) = \frac{1}{Z_P} W_P(y)\,\varphi_1(y - \mu_1^{\star}), \qquad Z_P = \int_{\mathbb{R}} W_P(t)\,\varphi_1(t - \mu_1^{\star})\,\mathrm{d}t. \tag{6}$$

We observe that $W_P(\cdot)$ is log-concave function.

**Lemma A.3.** $W_P(\cdot)$ *is log-concave.*

*Proof of Lemma A.3.* Consider the following density over $(y, z)$

$$\propto \mathbb{1}_P(y, z) \cdot \phi(z; \mu_{d-1}, I_{d-1})$$

This is log-concave because both $\mathbb{1}_P(y, z)$ and $\phi_z(\cdot)$ are log-concave; the former because $P$ is a convex set. The log-concavity of $W_P$ follows since it is a marginal of the above density on $y$, and log-concavity is preserved upon taking marginals. □

We will need the following one-dimensional variance reduction inequality and its equality case.

**Lemma A.4** (Lemma 4.8 in (Vempala, 2010)). *Let $Y \sim \mathcal{N}(\mu, \sigma^2)$ and $W : \mathbb{R} \to [0, \infty)$ be log-concave. Let $Y_W$ have density proportional to $W(y)\phi(y; \mu, \sigma^2)$. Then*

$$\operatorname{Var}(Y_W) \leq \sigma^2.$$

*Moreover, equality holds if and only if $\operatorname{supp} W = \mathbb{R}$ and $W$ is constant on $\mathbb{R}$.*

We note that Vempala (2010) proved the case for $\sigma = 1$, but the proof holds for any $\sigma > 0$. Moreover, they stated Lemma A.4 for the case $Y_W$ is centered. This is without loss of generality since the LHS variance is shift-invariant so we can consider the shifted function $W(\cdot + \mathbb{E}[Y_W])$ and shifted Gaussian $Y' \sim \mathcal{N}(\mu - \mathbb{E}[Y_W], \sigma^2)$. We now apply Lemma A.4 to each set in the full-mass sub-family given by Lemma A.2.

**Lemma A.5** (Constant slice weight for a.e. set). *Under the assumptions of Lemma A.2, for almost every $P \in \mathcal{P}$, there exists $c_P \in (0, 1]$ such that $W_P(y) \equiv c_P$ for all $y \in \mathbb{R}$.*

*Proof.* Fix such a $P$. By (6) and Lemma A.2, $\operatorname{Var}_{f_P}(y) = 1$. Apply Lemma A.4 with $W = W_P$: since $W_P$ is log-concave, equality in Lemma A.4 implies that $W_P$ is constant, and the support must be all of $\mathbb{R}$. Denote the resulting constant by $c_P$. □

Next, we show that the fact that $W_P(\cdot)$ being a constant forces the set $P$ to be a slab.

**Proposition A.6** (Constant sections $\Rightarrow$ slab). *Assume $P$ is convex and $W_P(\cdot) \equiv c_P > 0$. Then there exists a convex $C_P \subseteq u^\perp$ such that*

$$C_P(y) \equiv C_P \quad \text{for all } y \in \mathbb{R}, \qquad \text{and hence} \qquad P = \mathbb{R}u \oplus C_P,$$

i.e., *$P$ is a slab along $u$.*

To prove Proposition A.6, we use the following equality case of Prékopa–Leindler inequality due to (Ball & Böröczky, 2011; Dubuc, 1977).

**Theorem A.7** (Prékopa–Leindler Inequality, *e.g.*, Theorem 1.1 in Cordero-Erausquin & Maurey (2016)). *Let $f, g, h \colon \mathbb{R}^d \to [0, \infty)$ be measurable. Suppose that for some $t \in (0, 1)$ and for all $x, y \in \mathbb{R}^d$,*

$$h((1 - t)x + ty) \geq f(x)^{1-t} g(y)^t.$$

*Then*

$$\int_{\mathbb{R}^d} h(z) \, \mathrm{d}z \geq \left( \int_{\mathbb{R}^d} f(x) \, \mathrm{d}x \right)^{1-t} \left( \int_{\mathbb{R}^d} g(y) \, \mathrm{d}y \right)^t.$$

*Further, if equality holds above, then there exist $w \in \mathbb{R}^d$, $a > 0$, and a log-concave $\psi$ such that, a.e.,*

$$f(x) = a^{-t}\psi(x - tw), \qquad g(y) = a^{1-t}\psi(y + (1-t)w), \qquad \text{and} \qquad h(z) = \psi(z).$$

*Proof of Proposition A.6.* Fix $y_1, y_2 \in \mathbb{R}$, define $y_t := (1-t)y_1 + ty_2$, $A := C_P(y_1)$, $B := C_P(y_2)$, and $M_t := (1 - t)A + tB$.

First, because $P$ is convex, we can show the following.

**Lemma A.8.** $M_t \subseteq C_P(y_t)$.

*Proof.* To see this, consider any $z_1 \in C_P(y_1)$ and $z_2 \in C_P(y_2)$, then $(y_1, z_1), (y_2, z_2) \in P$; by convexity, $(y_t, (1-t)z_1 + tz_2) \in P$, *i.e.*, $(1-t)z_1 + tz_2 \in C_P(y_t)$. $\square$

Next, because $\varphi_{d-1}$ is log-concave, Prékopa–Leindler (Theorem A.7) yields the following:

**Lemma A.9.**

$$\int_{M_t} \varphi_{d-1} \geq \left[ \int_A \varphi_{d-1} \right]^{1-t} \left[ \int_B \varphi_{d-1} \right]^t.$$

*Proof.* To see this, let $f = \mathbb{1}_A \varphi_{d-1}$, $g = \mathbb{1}_B \varphi_{d-1}$ and $h = \mathbb{1}_{M_t} \varphi_{d-1}$ on $u^\perp$. For $x \in A, y \in B$ we have $(1 - t)x + ty \in M_t$, so $\mathbb{1}_{M_t}((1-t)x + ty) \geq \mathbb{1}_A(x)^{1-t}\mathbb{1}_B(y)^t$. Since $\varphi_{d-1}$ is log-concave, $\varphi_{d-1}((1-t)x + ty) \geq \varphi_{d-1}(x)^{1-t}\varphi_{d-1}(y)^t$. Multiplying gives the premise of Prékopa–Leindler inequality for $(f, g, h)$ (see Theorem A.7) and using the Prékopa–Leindler inequality yields the claim. $\square$

Combining Lemmas A.8 and A.9 with $\varphi_{d-1} > 0$ everywhere yields

$$\underbrace{\int_{C_P(y_t)} \varphi_{d-1}}_{= W_P(y_t) = c_P} \overset{\text{Lemma A.8}}{\geq} \int_{M_t} \varphi_{d-1} \overset{\text{Lemma A.9}}{\geq} \underbrace{\left( \int_{C_P(y_1)} \varphi_{d-1} \right)^{1-t} \left( \int_{C_P(y_2)} \varphi_{d-1} \right)^t}_{= c_P}. \quad (7)$$

(To see the last equality, note that $W_P(y) = \int_{C_P(y)} \varphi_{d-1}$ and $W_P(\cdot) \equiv c_P$.)

Since the leftmost and rightmost terms in (7) are equal, both intermediate inequalities are equalities. In particular, $C_P(y_t) \setminus M_t$ has Gaussian measure 0 (with respect to $\varphi_{d-1} \, \mathrm{d}z$).

Now we use the equality case of Prékopa–Leindler to conclude. By the equality case (Theorem A.7), there exist $w \in u^\perp$, $a > 0$, and a log-concave $\psi$ such that a.e. on $u^\perp$

$$f = \mathbb{1}_A \varphi_{d-1} = a^{-t} \psi(\cdot - tw), \qquad g = \mathbb{1}_B \varphi_{d-1} = a^{1-t} \psi(\cdot + (1-t)w), \qquad h = \mathbb{1}_{M_t} \varphi_{d-1} = \psi. \tag{8}$$

This, in particular, implies that $A = C + tw$ and $B = C - (1-t)w$ for $C := \mathrm{supp}(\psi)$ and, therefore,

$$M_t = (1-t)A + tB = (1-t)(C + tw) + t(C - (1-t)w) = C.$$

Hence, by (8), $\psi = \varphi_{d-1}$ on $C$ (a.e.). Therefore, for any $z \in C + tw = A$,

$$\varphi_{d-1}(z) = f(z) = a^{-t} \psi(z - tw) = a^{-t} \varphi_{d-1}(z - tw),$$

i.e., $\varphi_{d-1}(z) \equiv \kappa \, \varphi_{d-1}(z - tw)$ on $C + tw$ for $\kappa = a^{-t} > 0$. Hence, the map $z \mapsto \log \varphi_{d-1}(z) - \log \varphi_{d-1}(z - tw)$ must be a constant on $C$. However, by definition of $\varphi(\cdot)$

$$z \longmapsto \log \varphi_{d-1}(z) - \log \varphi_{d-1}(z - tw) = -\tfrac{1}{2}\Big(|z - \mu_\perp|^2 - |z - \mu_\perp - tw|^2\Big) = -t \, \langle z - \mu_\perp, w\rangle + \tfrac{t^2}{2} |w|^2$$

is affine with gradient $-t \, w$. Thus, if $w \neq 0$, it cannot be constant on any set with nonempty interior. But $C$ has nonempty interior: indeed $W_P(y_1) = \int_{C_P(y_1)} \varphi_{d-1} = c_P > 0$ and $C_P(y_1)$ is convex, hence it has positive Lebesgue measure and nonempty interior, so $\mathrm{int}(C) \neq \varnothing$. Therefore $w = 0$, and consequently

$$C_P(y_1) = C + tw = C = C - (1-t)w = C_P(y_2).$$

Since this holds for all $y_1, y_2$, there exists a convex $C_P \subseteq u^\perp$ with $C_P(y) \equiv C_P$ for all $y$. This implies the desired claim:

$$P = \{(y, z) : y \in \mathbb{R}, \ z \in C_P\} = \mathbb{R}u \oplus C_P. \qquad \square$$

Combining Lemma A.5 and Proposition A.6 with Lemma A.2 shows that almost every set $P$ is a slab along $u$; equivalently, the sub-family of non-slab sets has coarse mass 0.

### A.3 Converse: Parallel Slabs (Almost Everywhere) Are Non-Identifiable

Assume there exists $v \in \mathcal{S}^{d-1}$ and a sub-family $\mathcal{N} \subseteq \mathscr{P}$ with

$$\mathcal{N}_{\mathscr{P}}\left(\mu^\star, I\right)(\mathcal{N}) = 0 \qquad \text{such that} \qquad P \text{ is a slab along } v \text{ for every } P \in \mathscr{P} \setminus \mathcal{N}.$$

We show that $\mathcal{N}_{\mathscr{P}}(\mu, I) = \mathcal{N}_{\mathscr{P}}(\mu + tv, I)$ for all $\mu \in \mathbb{R}^d$ and $t \in \mathbb{R}$, hence identifiability fails.

Because the Gaussian densities $\phi_d(\cdot - \mu)$ are strictly positive for all $\mu$, the statement $\mathcal{N}_{\mathscr{P}}\left(\mu^\star, I\right)(\mathcal{N}) = 0$ implies that $\bigcup_{P \in \mathcal{N}} P$ has Lebesgue measure 0. In particular, $\mathcal{N}_{\mathscr{P}}(\mu, I)(\mathcal{N}) = 0$ for every $\mu$. Thus, intuitively, the sub-family $\mathcal{N}$ does not affect the population negative-log likelihood for any $\mu$. Hence, roughly speaking, checking non-identifiability for $\mathscr{P}$ reduces to checking non-identifiability for the sub-partition $\mathcal{S} := \mathscr{P} \setminus \mathcal{N}$ denoting the slab sub-family. For this sub-family non-identifiability easily follows due to symmetry along the "slab direction." In the remainder of the proof, we formally prove this.

Towards this, rotate coordinates so that $v = e_1$ and write $x = (y, z) \in \mathbb{R} \times v^\perp$ and $\mu = (\mu_v, \mu_\perp)$. For any slab cell $P = \mathbb{R} \times C_P$ with $C_P \subseteq v^\perp$ convex, the Gaussian factorizes:

$$\phi_d(x - \mu) = \phi_1(y - \mu_v) \, \phi_{d-1}(z - \mu_\perp).$$

Hence

$$\mathcal{N}_{\mathscr{P}}(\mu, I; P) = \int_{\mathbb{R}} \phi_1(y - \mu_v) \, dy \, \cdot \int_{C_P} \phi_{d-1}(z - \mu_\perp) \, \mathrm{d}z = \int_{C_P} \phi_{d-1}(z - \mu_\perp) \, \mathrm{d}z,$$

which is independent of $\mu_v$. Therefore, for every $t \in \mathbb{R}$,

$$\mathcal{N}_{\mathscr{P}}(\mu + tv, I; P) = \mathcal{N}_{\mathscr{P}}(\mu, I; P) \qquad \text{for all} \qquad P \in \mathscr{P} \setminus \mathcal{N}.$$

For any (measurable) collection $\mathcal{A} \subseteq \mathcal{P}$, set

$$U_{\mathcal{A}} := \bigcup_{P \in \mathcal{A} \cap \mathcal{S}} P \subseteq \mathbb{R}^d \qquad \text{and} \qquad N_{\mathcal{A}} := \bigcup_{P \in \mathcal{A} \cap n} P \,.$$

Then the Lebesgue measure of $N_{\mathcal{A}}$ is 0 and, by the product form of the slabs in $\mathcal{S}$,

$$U_{\mathcal{A}} = \mathbb{R}\, v \,\oplus\, \left( \bigcup_{P \in \mathcal{A} \cap \mathcal{S}} C_P \right) \,.$$

Let $\cup C_P$ denote $\bigcup_{P \in \mathcal{A} \cap \mathcal{S}} C_P$. Thus, for any $\mu$ and $t \in \mathbb{R}$,

$$\mathcal{N}_{\mathcal{P}}(\mu + tv, I)(\mathcal{A}) = \int_{U_{\mathcal{A}}} \phi_d(x - (\mu + tv))\, dx \;+\; \underbrace{\int_{N_{\mathcal{A}}} \phi_d(x - (\mu + tv))\, \mathrm{d}x}_{=\,0}$$

$$= \int_{\mathbb{R}} \phi_1[y - (\mu_v + t)]\, \mathrm{d}y \int_{\cup C_P} \phi_{d-1}[z - \mu_\perp]\, \mathrm{d}z$$

$$= \mathcal{N}_{\mathcal{P}}(\mu, I)(\mathcal{A}) \,,$$

since $\int_{\mathbb{R}} \phi_1(\cdot)\, dy = 1$. Because this holds for every $\mathcal{A}$, the pushforward measures coincide:

$$\mathcal{N}_{\mathcal{P}}(\mu + tv, I) \;=\; \mathcal{N}_{\mathcal{P}}(\mu, I) \quad \text{for all } t \in \mathbb{R} \,.$$

Therefore, shifting $\mu$ along $v$ leaves the coarse distribution and, hence, the negative log-likelihood unchanged, leading to the failure of identifiability.

## B   PROOF OF (THEOREM 3.2) (COARSE MEAN ESTIMATION ALGORITHM)

In this section, we provide our efficient algorithm for coarse Gaussian mean estimation under convex partitions. We begin by formalizing the input of the algorithm more generally as a sampling oracle, which can be implemented in polynomial time given a set of linear inequalities.

**Assumption 1** (Sampling Oracle). *There is an efficient sampling oracle that, given $R > 0$, $P \in \mathcal{P}$, and a parameter $\mu \in \mathbb{R}^d$ describing a Gaussian distribution, outputs an unbiased sample $y \sim \mathcal{N}(\mu, I, P \cap B_\infty(0, R))$.*

In light of Assumption 1, we now understand the input size as a natural measure of the complexity of the coarse observations (polytopes): the *facet-complexity*.[9] The running time of our algorithm will depend polynomially on the facet complexity of the observations. We note that this recovers the case when the coarse observations are encoded as a set of inequalities since the length of the encoding provides an upper bound on the facet complexity.

Next, we restate Theorem 3.2, the desired result, for convenience below.

**Theorem 3.2** (Coarse Mean Estimation Algorithm). *There is an algorithm that, for any true mean $\mu^\star \in \mathbb{R}^d$ with $\|\mu^\star\|_2 \leq D$ and any partition $\mathcal{P}$ that is convex and $\alpha$-information preserving with respect to $\mu^\star$ (Definition 3), gets as input the desired accuracy and confidence $\varepsilon, \delta \in (0, 1)$, draws i.i.d. coarse Gaussian samples from $\mathcal{P}$, and outputs an estimate $\widehat{\mu}$ satisfying*

$$\|\widehat{\mu} - \mu^\star\|_2 \leq \varepsilon$$

*with probability $1 - \delta$. The algorithms uses*

$$m = \widetilde{O}\left( \frac{dD^2 \log 1/\delta}{\alpha^4} + \frac{d \log 1/\delta}{\alpha^4 \varepsilon^2} \right)$$

*coarse samples and runs in time polynomial in $m$ and the bit complexity of the coarse samples.*

A few remarks about the assumptions in Theorem 3.2 are in order.

---

[9]The *facet-complexity* (Grötschel et al., 1988, Definition 6.2.2) of a polytope is a natural measure of the complexity of the polytope. See Appendix D.2 for the formal definition and more details.

**Remark B.1.** Assumption 1 can be further relaxed and is only stated for the sake of simplifying our presentation. For example, in the case where each $P \in \mathscr{P}$ is a polyhedron, we can remove Assumption 1 by implementing a sampling oracle that terminates in polynomial time with respect to the complexity of the input polyhedron. See Appendix D.1 about general convex bodies and Appendix D.2 for more polytope-specific details.

In addition to information preservation of the partition of $\mathbb{R}^d$ and the simplifying Assumption 1, our algorithm assumes the given partition is convex and that we are provided a bound on the norm of the true mean parameter. The convexity of the sets are crucial in order to argue about the convexity of the negative log-likelihood objective, but our algorithm is able to handle the case $\mu^\star \in B(\mu^{(0)}, D)$ via a translation of the space. Moreover, our algorithm can tolerate $\mu^\star \in B_\infty(\mu^{(0)}, D)$, *e.g.*, when we are provided a single fine sample $\mu^{(0)} \sim \mathcal{N}(\mu^\star, I)$, but for simplicity, we state our results using the Euclidean distance.

**Proof Overview** Our main workhorse is a variant of projected stochastic gradient descent (Theorem C.1). In order to apply call on this general result, we need a few ingredients.

- Analyzing the likelihood function over all $\mathbb{R}^d$ and for all partitions is challenging. Thus, we impose a feasible projection set and consider a slightly restricted class of convex partitions in Appendix B.1. The latter is used solely to facilitate analysis, and our final algorithm holds for general convex partitions.

- In order to show that the iterates of gradient descent converges in Euclidean distance and not just in function value, we show in Appendix B.2 that our function is convex and satisfies a local growth condition. This is a strictly weaker condition than strong convexity.

- Next, to bound the number of gradient steps, we design a stochastic gradient oracle and bound its second moment in Appendix B.3. Here, we crucially use properties of the local partition.

- Appendix B.4 collects the ingredients we discussed above to derive an algorithm for the afore-mentioned restricted convex partitions.

- Finally, Appendix B.5 reduces the case of general partitions to the case of restricted partitions.

- Appendix B.6 fills in the gradient and Hessian calculations that were deferred in this section.

## B.1 PROJECTION SET & LOCAL PARTITIONS

To be able to run PSGD on the negative log-likelihood, we need to obtain stochastic gradients for $\mathscr{L}_{\mathscr{P}}$ as well as bound the second moment of the stochastic gradients. This is not straightforward since the inner and outer expectations in the gradient expressions (Appendix B.6) are over different mean parameters.

To facilitate the analysis, we first impose a projection set $K$, over which we can ensure a trivial bound on the distance of the current iterate to the true solution. Our projection set $K$ is defined to be

$$K := \left\{ \mu \in \mathbb{R}^d \colon \|\mu\|_2 \leq D \right\}, \tag{9}$$

where $D$ is the assumed bound on the norm of the true mean. We certainly have $\mu^\star \in K$ and we can efficiently project onto $K$.

Furthermore, we first analyze an idealized class of partitions and derive an algorithm that recovers the mean under this ideal situation. Then we show that we can implement this algorithm using samples from the actual class of observed partitions.

**Definition 5** ($R$-Local Partition). *Let $R > 0$. We say a partition $\mathscr{P}$ of $\mathbb{R}^d$ is $R$-local if for any $P \in \mathscr{P}$ that is not a singleton, $P \subseteq B_\infty(0, R)$.*

## B.2 Convexity and Local Growth of Log-Likelihood Under Convex Partitions

It can be verified (Appendix B.6) that the coarse negative log-likelihood of the mean under canonical parameterization is given by the following

$$\mathscr{L}_{\mathscr{P}}(\mu) = \sum_{P \in \mathscr{P}} \mathcal{N}(\mu^\star, I; P) \, \mathscr{L}_P(\mu), \quad \text{where} \quad \mathscr{L}_P(\mu) := -\log\left(\mathcal{N}(\mu, I; P)\right). \tag{10}$$

We also have the following expressions for the gradient and Hessian of $\mathscr{L}_{\mathscr{P}}$.

$$\boldsymbol{\nabla}\mathscr{L}_{\mathscr{P}}(\mu) = \mu - \underset{P \sim \mathcal{N}_{\mathscr{P}}(\mu^\star, I)}{\mathbb{E}} \underset{\mathcal{N}(\mu, I, P)}{\mathbb{E}} [x],$$

$$\boldsymbol{\nabla}^2\mathscr{L}_{\mathscr{P}}(\mu) = I - \underset{P \sim \mathcal{N}_{\mathscr{P}}(\mu^\star, I)}{\mathbb{E}} \underset{\mathcal{N}(\mu, I, P)}{\mathrm{Cov}} [x].$$

Fotakis et al. (2021) show that $\boldsymbol{\nabla}^2\mathscr{L}_{\mathscr{P}}(\mu) \succeq 0$ when each $P \in \mathscr{P}$ is convex by leveraging a variance reduction inequality, which is implied by the Brascamp–Lieb inequality (see *e.g.*, Guionnet (2009); Hargé (2004)) and showing that each $\mathscr{L}_P$ is convex.

**Proposition B.2** (Lemma 15 in Fotakis et al. (2021)). *Let $P \subseteq \mathbb{R}^d$ be convex. Then $\mathrm{Cov}_{\mathcal{N}(\mu, I, P)}[x] \preceq I$, and in consequence, $\mathscr{L}_P(\mu)$ is convex as a function of $\mu$.*

To recover $\mu^\star$, one sufficient condition is to check that $\mathscr{L}_{\mathscr{P}}$ satisfies a local growth condition (Definition 6). We show this in the next lemma via a connection between the log-likelihood objective and information-preservation of $\mathscr{P}$.

**Lemma B.3.** *Let $\mathscr{P}$ be a convex $\alpha$-information preserving partition of $\mathbb{R}^d$. The negative log-likelihood function $\mathscr{L}_{\mathscr{P}}(\mu)$ (Equation (10)) satisfies a $\left(\frac{\sqrt{2}\alpha}{200}, \frac{2\alpha^2}{600^2}\right)$-local growth condition.*

We note that a similar statement is proven by Kalavasis et al. (2025). However, their definition of information preservation (Definition 3) is slightly different in that they directly assume the coarse total-variation distance is lower bounded by the parameter distance. Thus, their analysis skips the need to relate non-coarse Gaussian total-variation to parameter distances which is necessary for us.

*Proof.* Assume that $\alpha$-information preservation (Definition 3) holds. The proof relies on a tight characterization between the TV distance of a Gaussian and its parameters due to Arbas et al. (2023, Theorem 1.8): For any $\mu_1, \mu_2 \in \mathbb{R}^d$ such that $\mathrm{d}_{\mathsf{TV}}(\mathcal{N}(\mu_1, I), \mathcal{N}(\mu_2, I)) < 1/600$,

$$\frac{1}{200} \|\mu_1 - \mu_2\|_2 \leq \mathrm{d}_{\mathsf{TV}}(\mathcal{N}(\mu_1, I), \mathcal{N}(\mu_2, I)) \leq \frac{1}{\sqrt{2}} \|\mu_1 - \mu_2\|_2.$$

For any $\mu \in \mathbb{R}^d$ such that $\mathscr{L}(\mu) - \mathscr{L}(\mu^\star) < \frac{2\alpha^2}{600^2}$, we have

$$\mathrm{d}_{\mathsf{TV}}(\mathcal{N}(\mu, I), \mathcal{N}(\mu^\star, I)) \leq \frac{1}{\alpha} \mathrm{d}_{\mathsf{TV}}(\mathcal{N}_{\mathscr{P}}(\mu, I), \mathcal{N}_{\mathscr{P}}(\mu^\star, I))$$

$$\leq \frac{1}{\alpha} \sqrt{\frac{1}{2} \mathsf{KL}(\mathcal{N}_{\mathscr{P}}(\mu, I) \| \mathcal{N}_{\mathscr{P}}(\mu^\star, I))} \qquad \text{(Pinsker's inequality)}$$

$$= \frac{1}{\alpha} \sqrt{\frac{1}{2} [\mathscr{L}(\mu) - \mathscr{L}(\mu^\star)]}$$

$$< \frac{1}{600}.$$

It follows that the characterization above applies and we can lower bound the KL-divergence using the parameter distance:

$$\mathscr{L}(\mu) - \mathscr{L}(\mu^\star) = \mathsf{KL}(\mathcal{N}_{\mathscr{P}}(\mu, I) \| \mathcal{N}_{\mathscr{P}}(\mu^\star, I)) \qquad \text{(Pinsker's inequality)}$$

$$\geq 2 \cdot \mathrm{d}_{\mathsf{TV}}(\mathcal{N}_{\mathscr{P}}(\mu, I), \mathcal{N}_{\mathscr{P}}(\mu^\star, I))^2$$

$$\geq 2\alpha^2 \cdot \mathrm{d}_{\mathsf{TV}}(\mathcal{N}(\mu, I), \mathcal{N}(\mu^\star, I))^2$$

$$\geq \frac{2\alpha^2}{200^2} \|\mu - \mu^\star\|_2^2. \qquad \square$$

### B.3 Bounding the Second Moment of Stochastic Gradients for Local Partitions

We will prove the bound on the second moment of the stochastic gradient for an $R$-local partition. We will then show how to deal with a general partition.

**Lemma B.4.** *Suppose $\mathscr{P}$ is an $R$-local partition of $\mathbb{R}^d$ and $\|\mu^\star\|_2 \leq D$. Given $v \in K$ in the projection set (Equation (9)), there is an algorithm that consumes a sample $P \sim \mathcal{N}_\mathscr{P}(\mu^\star, I)$ and makes a single call to the sampling oracle (Assumption 1) to compute an unbiased estimate $g(\mu)$ of $\nabla \mathscr{L}_\mathscr{P}(\mu)$ (Equation (10)) such that $\mathbb{E}\left[\|g(\mu)\|_2^2\right] = O(D^2 + dR^2)$.*

Recall that the gradient of the log-likelihood function is of the form

$$\mu - \mathop{\mathbb{E}}_{P \sim \mathcal{N}_\mathscr{P}(\mu^\star, I)} \mathop{\mathbb{E}}_{\mathcal{N}(\mu, I, P)} [x].$$

*Proof.* The first term in the gradient expression is simply our current iterate for the mean and we can obtain an unbiased estimate of the second term by sampling $y \sim \mathcal{N}(\mu, I, P)$ where $P$ is a fresh observation.

The second moment of the gradient oracle can be upper bounded by

$$2\|\mu\|_2^2 + 2 \mathop{\mathbb{E}}_{P \sim \mathcal{N}_\mathscr{P}(\mu^\star, I)} \mathop{\mathbb{E}}_{\mathcal{N}(\mu, I, P)} \left[\|x\|_2^2\right].$$

The first term is at most $D^2$ and the second term can be upper bounded as follows.

$$\mathop{\mathbb{E}}_{P \sim \mathcal{N}_\mathscr{P}(\mu^\star, I)} \mathop{\mathbb{E}}_{\mathcal{N}(\mu, I, P)} \left[\|x\|_2^2\right]$$

$$= \mathop{\mathbb{E}}_{P \sim \mathcal{N}_\mathscr{P}(\mu^\star, I)} \mathop{\mathbb{E}}_{x \sim \mathcal{N}(\mu, I, P)} \left[\|x\|_2^2 \cdot \chi_{B_\infty(0, R)}\right] + \mathop{\mathbb{E}}_{P \sim \mathcal{N}_\mathscr{P}(\mu^\star, I)} \mathop{\mathbb{E}}_{x \sim \mathcal{N}(\mu, I, P)} \left[\|x\|_2^2 \cdot \chi_{B_\infty(0, R)^c}\right]$$

$$= \mathop{\mathbb{E}}_{P \sim \mathcal{N}_\mathscr{P}(\mu^\star, I)} \mathop{\mathbb{E}}_{x \sim \mathcal{N}(\mu, I, P)} \left[\|x\|_2^2 \cdot \chi_{B_\infty(0, R)}\right] + \mathop{\mathbb{E}}_{P \sim \mathcal{N}_\mathscr{P}(\mu^\star, I)} \mathop{\mathbb{E}}_{x \sim \mathcal{N}(\mu^\star, I, P)} \left[\|x\|_2^2 \cdot \chi_{B_\infty(0, R)^c}\right].$$

In the last step, we use the fact that $\mathscr{P}$ is an $R$-local partition so that every set such that $P \cap B_\infty(0, R)^c \neq \varnothing$ is a singleton. This allows us to replace the expectation over each $\mathcal{N}(\mu, I, P)$ in the second term with an expectation over $\mathcal{N}(\mu^\star, I, P)$, as the distribution consists of a single point. We can then bound the first term using a deterministic bound and the second term using the second moment of $\mathcal{N}(\mu^\star, I)$.

$$\mathop{\mathbb{E}}_{P \sim \mathcal{N}_\mathscr{P}(\mu^\star, I)} \mathop{\mathbb{E}}_{x \sim \mathcal{N}(\mu, I, P)} \left[\|x\|_2^2 \cdot \chi_{B_\infty(0, R)}\right] + \mathop{\mathbb{E}}_{P \sim \mathcal{N}_\mathscr{P}(\mu^\star, I)} \mathop{\mathbb{E}}_{x \sim \mathcal{N}(\mu^\star, I, P)} \left[\|x\|_2^2 \cdot \chi_{B_\infty(0, R)^c}\right]$$

$$\leq dR^2 + \mathop{\mathbb{E}}_{x \sim \mathcal{N}(\mu^\star, I)} \left[\|x\|_2^2\right]$$

$$= dR^2 + d + \|\mu^\star\|_2^2$$

$$= O(dR^2 + D^2).$$

$\square$

### B.4 Projected Stochastic Gradient Descent for Local Partitions

Having ensured a bound on the second moment for a local partition, we can now run the iterative PSGD algorithm for functions satisfying local growth (Theorem C.1). Applying its analysis with the ingredients above yields the following result.

**Proposition B.5.** *Let $\varepsilon \in (0, 1)$. Suppose $\mathscr{P}$ is a convex $R$-local $\alpha$-information preserving partition of $\mathbb{R}^d$ and $\|\mu^\star\|_2 \leq D$. There is an algorithm that outputs an estimate $\widetilde{\mu}$ satisfying*

$$\|\widetilde{\mu} - \mu^\star\|_2 \leq \varepsilon$$

*with probability $1 - \delta$. Moreover, the algorithm requires*

$$m = O\left(\frac{dR^2 + D^2}{\alpha^4 \varepsilon^2} \log^3\left(\frac{dRD}{\varepsilon\delta}\right)\right)$$

*samples from $\mathcal{N}_{\mathscr{P}}(\mu^\star, I)$ and $\mathrm{poly}(m, T_s)$ time, where $T_s$ is the time complexity of sampling from a Gaussian distribution truncated to $P \cap B_\infty(0, R)$ for some $P \in \mathscr{P}$.*

*Proof.* We call on Theorem C.1. The initial function value bound can be taken as $\varepsilon_0 = DG$ with any initial value $\mu_0 \in B(0, D)$, where $G^2 = O(D^2 + dR^2)$ is an upper bound on the second moment of the gradient oracle (Lemma B.4). In order to recover $\mu^\star$ up to $\varepsilon$-Euclidean distance, we would like an $O(\alpha^2 \varepsilon^2)$-optimal solution in function value. In order to obtain the high probability bound, we repeat Theorem C.1 a few times and apply the clustering trick described in Remark C.2. $\qquad\square$

## B.5 Reducing General Partitions to Local Partitions

We now describe how to remove the assumption of the partition being $R$-local and prove Theorem 3.2.

*Proof of Theorem 3.2.* Consider a general $\alpha$-information preserving convex partition $\mathscr{P}$ and $P \sim \mathcal{N}_{\mathscr{P}}(\mu^\star, I)$. Since $\|\mu^\star\|_\infty \leq \|\mu^\star\|_2 \leq D$, setting $R = D + O(\log^{md}/\delta)$ means that any $m$-sample algorithm will not observe a sample $P$ such that $P \cap B_\infty(0, R) = \varnothing$ with probability $1 - \delta$. Define the partition $\mathscr{P}(R)$ of $\mathbb{R}^d$ given by

$$\mathscr{P}(R) \coloneqq \{P \cap B_\infty(0, R) : P \in \mathscr{P}, P \cap B_\infty(0, R) \neq \varnothing\} \cup \{\{x\} : x \notin B_\infty(0, R)\} .$$

Since $\mathscr{P}(R)$ is a refinement of $\mathscr{P}$, it must also be $\alpha$-information preserving. Consider the set-valued algorithm $F \colon \mathscr{P} \to \mathscr{P}(R)$ given by

$$P \mapsto \begin{cases} P \cap B_\infty(0, R), & P \cap B_\infty(0, R) \neq \varnothing , \\ \{x\} \text{ for an arbitrary } x \in P, & P \cap B_\infty(0, R) = \varnothing . \end{cases}$$

By the choice of $R$, with probability $1 - \delta$, any $m$-sample algorithm will not distinguish between samples from $\mathcal{N}_{\mathscr{P}(R)}(\mu^\star, I)$ and $F(P)$ for $P \sim \mathcal{N}_{\mathscr{P}}(\mu^\star, I)$. Moreover, we can sample from $P \cap B_\infty(0, R)$ as per Assumption 1. Thus we can run the algorithm from Proposition B.5 with input samples $F(P)$ for $P \sim \mathcal{N}_{\mathscr{P}}(\mu^\star, I)$. $\qquad\square$

We can further improve the sample complexity by running the algorithm in two stages. The first stage aims to obtain an $O(1)$-distance warm start and the second stage aims to recover the mean up to accuracy $\varepsilon$. This yields the proof of Theorem 3.2. Indeed, the only missing detail is an efficient implementation of a sampling oracle (Assumption 1). As mentioned, we defer these details to Appendix D.2.

## B.6 Gradient & Hessian Computations

In this section, we fill in the gradient and Hessian calculations for the log-likelihood function for coarse mean estimation under a general convex partition. Let $\mathscr{L} \colon \mathbb{R}^d \to \mathbb{R}_{\geq 0}$ denote the negative log-likelihood function for an instance of the coarse Gaussian mean estimation problem. Similar calculations are presented by Fotakis et al. (2021); Kalavasis et al. (2025), but we include a self-contained proof for completeness. $\mathscr{L}$ is defined as follows

$$\mathscr{L}(\mu) \coloneqq \mathop{\mathbb{E}}_{P \sim \mathcal{N}_{\mathscr{P}}(\mu^\star, I)} \left[-\log\left(\mathcal{N}(\mu, I; P)\right)\right] .$$

We first compute the function derivatives.

**Fact B.6.** *It holds that*

$$\boldsymbol{\nabla}\mathscr{L}(\mu) = \mathop{\mathbb{E}}_{\mathcal{N}(\mu, I)}[x] - \mathop{\mathbb{E}}_{P \sim \mathcal{N}_{\mathscr{P}}(\mu^\star, I)} \mathop{\mathbb{E}}_{\mathcal{N}(\mu, I, P)}[x] ,$$

$$\boldsymbol{\nabla}^2\mathscr{L}(\mu) = \mathop{\mathrm{Cov}}_{\mathcal{N}(\mu, I)}[x] - \mathop{\mathbb{E}}_{P \sim \mathcal{N}_{\mathscr{P}}(\mu^\star, I)} \mathop{\mathrm{Cov}}_{\mathcal{N}(\mu, I, P)}[x] ,$$

*Proof.* We can write the log-likelihood as follows

$$\mathscr{L}(\mu) = \sum_{P \in \mathscr{P}} \mathcal{N}(\mu^\star, I; P) \mathscr{L}_P(\mu), \quad \text{where} \quad \mathscr{L}_P(\mu) := -\log\left(\mathcal{N}(\mu, I; P)\right). \tag{11}$$

Due to the linearity of gradients, it suffices to compute $\boldsymbol{\nabla}\mathscr{L}_P(v, T)$ and $\boldsymbol{\nabla}^2\mathscr{L}_P(\mu)$ for each $P \in \mathscr{P}$ to obtain the gradient and Hessian of $\mathscr{L}(\cdot)$. Toward this, fix any $P \in \mathscr{P}$. Observe that

$$\mathscr{L}_P(\mu) = \log \frac{\int_{x \in \mathbb{R}^d} e^{-\frac{1}{2}(x-\mu)^\top (x-\mu)} \mathrm{d}x}{\int_{x \in P} e^{-\frac{1}{2}(x-\mu)^\top (x-\mu)} \mathrm{d}x} = \log \frac{\int_{x \in \mathbb{R}^d} e^{-\frac{1}{2}\|x\|_2^2 + x^\top \mu} \mathrm{d}x}{\int_{x \in P} e^{-\frac{1}{2}\|x\|_2^2 + x^\top \mu} \mathrm{d}x}.$$

Write $f(x; \mu) := \exp\left(-\frac{1}{2}\|x\|_2^2 + x^\top \mu\right)$. It follows that

$$\begin{aligned}
\boldsymbol{\nabla}_\mu \mathscr{L}_P(\mu) &= \frac{\int_P f}{\int_{\mathbb{R}^d} f} \cdot \left[\frac{(\int_{\mathbb{R}^d} \boldsymbol{\nabla}f)(\int_P f) - (\int_{\mathbb{R}^d} f)(\int_P \boldsymbol{\nabla}f)}{(\int_P f)^2}\right] \\
&= \frac{\int_{\mathbb{R}^d} \boldsymbol{\nabla}f}{\int_{\mathbb{R}^d} f} - \frac{\int_P \boldsymbol{\nabla}f}{\int_P f}.
\end{aligned} \tag{12}$$

Simplifying the expression and substituting the values of $f, \boldsymbol{\nabla}f$ gives

$$\boldsymbol{\nabla}\mathscr{L}_P(\mu) = \mathop{\mathbb{E}}_{\mathcal{N}(\mu, I)}[x] - \mathop{\mathbb{E}}_{\mathcal{N}(P, \mu, I)}[x].$$

Substituting this in Equation (11) gives the desired expression for $\boldsymbol{\nabla}\mathscr{L}(v, T)$.

To compute $\boldsymbol{\nabla}^2\mathscr{L}_P(\mu)$, we differentiate Equation (12).

$$\begin{aligned}
\boldsymbol{\nabla}_\mu^2 \mathscr{L}_P(\mu) &= \frac{(\int_{\mathbb{R}^d} \boldsymbol{\nabla}^2 f)(\int_{\mathbb{R}^d} f) - (\int_{\mathbb{R}^d} \boldsymbol{\nabla}f)(\int_{\mathbb{R}^d} \boldsymbol{\nabla}f)^\top}{(\int_{\mathbb{R}^d} f)^2} - \frac{(\int_P \boldsymbol{\nabla}^2 f)(\int_P f) - (\int_P \boldsymbol{\nabla}f)(\int_P \boldsymbol{\nabla}f)^\top}{(\int_P f)^2} \\
&= \frac{\int_{\mathbb{R}^d} \boldsymbol{\nabla}^2 f}{\int_{\mathbb{R}^d} f} - \frac{(\int_{\mathbb{R}^d} \boldsymbol{\nabla}f)(\int_{\mathbb{R}^d} \boldsymbol{\nabla}f)^\top}{(\int_{\mathbb{R}^d} f)^2} \qquad - \left[\frac{\int_P \boldsymbol{\nabla}^2 f}{\int_P f} - \frac{(\int_P \boldsymbol{\nabla}f)(\int_P \boldsymbol{\nabla}f)^\top}{(\int_P f)^2}\right].
\end{aligned}$$

This simplifies to the following

$$\boldsymbol{\nabla}^2 \mathscr{L}_P(\mu) = \mathop{\mathbb{E}}_{\mathcal{N}(\mu, I)}\left[xx^\top\right] - \mathop{\mathbb{E}}_{\mathcal{N}(\mu, I)}[x] \mathop{\mathbb{E}}_{\mathcal{N}(\mu, I)}[x]^\top - \mathop{\mathbb{E}}_{\mathcal{N}(P, \mu, I)}\left[xx^\top\right] + \mathop{\mathbb{E}}_{\mathcal{N}(P, \mu, I)}[x] \mathop{\mathbb{E}}_{\mathcal{N}(P, \mu, I)}[x]^\top.$$

A final simplification gives

$$\boldsymbol{\nabla}^2 \mathscr{L}_P(\mu) = \mathop{\mathrm{Cov}}_{\mathcal{N}(\mu, I)}[x] - \mathop{\mathrm{Cov}}_{\mathcal{N}(P, \mu, I)}[x]$$

Substituting this in Equation (11) gives the desired expression of $\boldsymbol{\nabla}^2\mathscr{L}(\mu)$. $\qquad\square$

## C  PSGD Convergence for Convex Functions Satisfying Local Growth

In this section, we state the variant of gradient descent we use to optimize the various likelihood functions in our work. In order to establish the sample complexity and computational efficiency of our main algorithms (Theorem 3.2 and Theorem 3.3), we require convergence guarantees for projected stochastic gradient descent (PSGD) over functions that lack global strong convexity. To address this, we leverage a *local growth condition* (Definition 6), which our coarse negative log-likelihood functions satisfy under appropriate information-preservation assumptions. We subsequently present an iterative PSGD convergence statement (Theorem C.1) for convex functions endowed with this local growth property.

Before formally doing so, we state some useful definitions. Consider a convex function $F: K \to \mathbb{R}$ with a global minimizer $w^\star$ on a convex subset $K \subseteq \mathbb{R}^d$. We write

$$S_\rho := \{w \in K : F(w) - F(w^\star) \le \rho\}$$

to denote the *$\varepsilon$-sublevel set* of a function $F$ where $F$ is clear from context.

**Definition 6** ($\eta$-Local Growth Condition). *We say that $F\colon K \to \mathbb{R}$ satisfies an $(\eta, \rho)$-local growth condition if*

$$\|w - w^\star\|_2 \leq \frac{(F(w) - F(w^\star))^{\frac{1}{2}}}{\eta}$$

*for every $w \in S_\rho$.*

We remark that a function satisfying a $(\eta_0, \rho_0)$-local growth condition also satisfies $(\eta, \rho)$-local growth for every $\eta \in (0, \eta_0], \rho \in (0, \rho_0]$. We also note that our log-likelihood function for $\alpha$-information preserving partitions satisfy a $(\frac{\sqrt{2}\alpha}{200}, \frac{2\alpha^2}{600^2})$-local growth condition (Lemma B.3).

Our efficient coarse mean estimation algorithm uses the following result due to Kalavasis et al. (2025).

**Theorem C.1.** *Let $F\colon S \subseteq \mathbb{R}^d \to \mathbb{R}$ be convex and satisfy a $(\eta, \varepsilon)$-local growth condition (Definition 6). Suppose we have acccess to an unbiased gradient oracle $g(w)$ such that $\mathbb{E}\left[\|g(w)\|_2^2\right] \leq G^2$. Suppose further that we have access to an $\varepsilon_0$-optimal solution $w^{(0)} \in S$. There is an algorithm that queries the gradient oracle*

$$O\left(\frac{G^2}{\eta^2 \varepsilon} \cdot \log^3\left(\frac{\varepsilon_0}{\varepsilon}\right)\right)$$

*times and outputs a $2\varepsilon$-optimal solution with probability $0.99$.*

**Remark C.2** (Boosting without Function Evaluation via Clustering). For a function satisfying a $(\eta, \varepsilon)$ local growth condition, it suffices to apply Theorem C.1 and compute a $O(\eta^2 \varepsilon^2)$-optimal solution $w$ in function value to ensures that $\|w - w^\star\|_2 \leq \varepsilon$. Note that this procedure succeeds with probability $0.99$.

In order to boost the probability of successfully recovering $w^\star$, a standard trick is to repeat the algorithm $O(\log(1/\delta))$ times and output the solution with smallest objective value. However, we do not have exact evaluation access to the log-likelihood functions we wish to optimize. Daskalakis et al. (2018, Section 3.4.5) demonstrate a "clustering" trick to avoid function evaluation. Suppose we repeat the algorithm in Theorem C.1 $O(\log(1/\delta))$ times. A Chernoff bound yields that with high probability, at least $2/3$ of the outputted points are $\varepsilon$-close to $w^\star$ and thus are $2\varepsilon$-close to each other. Thus, outputting any point which is at most $2\varepsilon$-close to at least 50% of the points must be at most $3\varepsilon$-close to $w^\star$.

## D  LOG-CONCAVE SAMPLING OVER CONVEX BODIES

In this section, we review well-known results about sampling from a log-concave density $\propto e^{-f}$ constrained to a convex body $K$, under mild assumptions. We use these results to implement the sampling oracle (Assumption 1) in Appendix B and the friction sampling oracle (Assumption 7) in Appendix E.

Our stochastic gradient descent algorithms for coarse mean estimation (Theorem 3.2) and linear regression with friction (Theorem 3.3) rely on samples from truncated Gaussian distributions to compute stochastic gradients (Appendix B.3 and Appendix E.3). Here, we provide the technical statements that justify these algorithmic sampling oracles. Specifically, we first state a general hit-and-run sampling result for arbitrary convex bodies (Theorem D.1). We then show how to instantiate this general result to remove our idealized sampling assumptions for closed convex sets (Appendix D.1) and polyhedra (Appendix D.2), thereby fully validating the computational efficiency of our primary estimation algorithms.

Concretely, we have a convex function $f : K \to \mathbb{R}$ that is bounded below, say there is some $x^\star \in K$ such that $f(x) \geq f(x^\star)$ for all $x \in K$. Note that $x^\star$ exists and belongs in $K$ since $K$ is closed.

One option for this task is the class of Langevin Monte Carlo methods (Brosse et al., 2017; Bubeck et al., 2018; Dalalyan & Karagulyan, 2019; Lamperski, 2021; Ahn & Chewi, 2021), which requires first-order gradient access to the log-density function. A more general tool for this purpose is the "Hit-And-Run" Markov Chain Monte Carlo (MCMC) algorithm due to Lovász & Vempala (2006a), which only requires minimal zero-order access.

**Theorem D.1** (Mixing-Time of Hit-And-Run Markov Chains; Lovász & Vempala (2006b;a))**.** *Consider a logconcave distribution $\pi_f \propto e^{-f}$ over a convex body $K$. Suppose we are provided the following.*

*(S1) Zero-order access to $f$.*

*(S2) Membership access to $K$.*

*(S3) A point $x^{(0)}$ and constants $R \geq r > 0$ such that $B_2(x^{(0)}, r) \subseteq K \subseteq B_2(x^0, R)$.*

*(S4) A bound $M \geq \max_{x \in K} f(x) - f(x^\star)$.*

*Then, there is an algorithm that makes*

$$O\left(d^{4.5} \cdot \mathrm{polylog}(d, M, {}^{R}/{}_{r}, {}^{1}/{}_{\delta})\right)$$

*membership oracle calls to produce a random vector within $\delta$-TV distance of $\pi_f$.*

To implement the desired Markov chain, we need Assumptions (S1) to (S4) stated in Theorem D.1. We remark that Assumption (S2) and Assumption (S3) can both be implemented given a separation oracle to $K$. If we also know that $f$ is $L$-Lipschitz or $\beta$-smooth, we can take $M = LR$ or $M = \beta R^2$, respectively, for Assumption (S4). The smallest ratio ${}^{R}/{}_{r}$ of values $r, R$ from Assumption (S3) depends on the structure of $K$. For the simple case of 1-dimensional intervals, we trivially have ${}^{R}/{}_{r} = 1$. For all $d$-dimensional $L_p$-balls, we have ${}^{R}/{}_{r} = \mathrm{poly}(d)$.

## D.1 REMOVING ASSUMPTION 1 FOR CLOSED CONVEX SETS

For partitions containing general closed convex sets, the sampling oracle assumption (Assumption 1) from our coarse Gaussian mean estimation algorithm (Appendix B) can be relaxed to Assumption 2 below.

**Assumption 2** (Well-Bounded Set)**.** *Each set $P \in \mathscr{P}$ is a closed convex set and is encoded as:*

*(i) the dimension $k = \dim(P)$ of the affine hull of $P$,*

*(ii) a separation oracle,*

*(iii) numbers $R, r > 0$ with the promise that if $P \cap B_\infty^{(k)}(0, R) \neq \varnothing$, then $P$ contains an inner ball $P \supseteq B_2^{(k)}(x, r)$ for some $x \in B_\infty^{(k)}(0, R)$.*

*Here $B_p^{(k)}(\cdot, \cdot)$ denotes the $k$-dimensional $L_p$-ball with given center and radius.*

Assumption 2 allows for unbounded convex sets while ensuring that we can call on Theorem D.1 with the bounded convex body $K = P \cap B_\infty^{(k)}(0, R + r\sqrt{d})$, which contains most of the mass of $P$ for sufficiently large $R$. This is because we are guaranteed the existence of an (unknown) inner ball $B_2(x^{(0)}, r) \subseteq K$. We can further approximately find such a ball using the ellipsoid method (Grötschel et al., 1988). The overall running time to obtain an $\delta$-approximate sample in total variation distance is thus

$$\mathrm{poly}\left(k, \log({}^{R}/{}_{r}), \log({}^{1}/{}_{\delta})\right) .$$

In our mean estimation algorithm (Theorem 3.2), we may wish to sample from the set $P \cap B_\infty(0, R')$ for some other $R' < R$ sufficiently large. This can be implemented via rejection sampling from $P \cap B_\infty(0, R)$ assuming the mass of $B_\infty(0, R')$ is sufficiently large.

## D.2 REMOVING ASSUMPTION 1 FOR POLYHEDRA

In the case the partition consists of polyhedra, the sampling oracle assumption (Assumption 1) from our coarse Gaussian mean estimation algorithm (Appendix B) can be relaxed to Assumption 3 below. Before stating the exact assumption, we state a natural notion of the complexity of a polytope.

**Definition 7** (Facet-Complexity)**.** *We say that a polytope $P \subseteq \mathbb{R}^d$ has facet-complexity at most $\varphi_P$ if there exists a system of inequalities with rational coefficients that has solution set $P$ and such that the bit-encoding length of each inequality of the system is at most $\varphi_P$. In case $P = \mathbb{R}^d$, we require $\varphi_P \geq d + 1$.*

We are now ready to replace Assumption 1.

**Assumption 3** (Well-Described Polyhedron). *Each observation $P \in \mathscr{P}$ is a polyhedron and is provided in the form of a separation oracle and an upper bound $\varphi_P > 0$ on the facet complexity of $P$.*

The running time of our sampling algorithm then depends polynomially on the running time of the separation oracle as well as the facet-complexity $\varphi_P$ of the observed samples $P$. Indeed, let us check that Assumption 3 allows us to apply Theorem D.1. It is clear that Assumption 3 immediately satisfies Assumptions (S1) and (S2). We sketch how to address Assumptions (S3) and (S4).

First, note that we wish to sample from the truncated standard Gaussian $\mathcal{N}(\mu, I, P \cap B_\infty(0, R))$. In particular, it has density that is $O(1)$-smooth. Since $R = \text{poly}(d, D, \log(1/\delta))$ is a polynomial of the dimension $d$, the warm-start radius $D$, and logarithmic in the inverse failure probability $\delta$. we can take $M = O(R^2)$ so that Assumption (S4) is satisfied.

Next, we can again use the fact that we sample from polytopes contained in $B_\infty(0, R)$ to deduce that $P \cap B_\infty(0, R)$ is contained in an $L_2$ ball of radius $R\sqrt{d}$. Moreover, the facet-complexity of $P \cap B_\infty(0, R)$ is at most $\varphi = \varphi_P + \log_2(R)$. On the other hand, to handle the inner ball, we draw on (Grötschel et al., 1988, Lemma 6.2.5) which states that a full-dimensional polytope with facet-complexity $\varphi$ must contain a ball of radius $2^{-7d^3\varphi}$. In the case that $P \cap B_\infty(0, R)$ is full-dimensional, this suffices to run the Markov chain from Theorem D.1 in $\text{poly}(d, \varphi)$ time. If $P$ is not full-dimensional, we can exactly compute the affine hull in polynomial time using the ellipsoid algorithm (Grötschel et al., 1988, (6.1.2)) and then apply the full-dimensional argument on this affine subspace.

# E    ALGORITHM FOR LINEAR REGRESSION WITH FRICTION

In this section, we prove the main result on linear regression with friction (Theorem E.1). Before presenting the result, we define the formal model and state our assumptions.

**Definition 8** (Linear Regression with Friction). *Let $c : \mathbb{R} \to \mathbb{R}$ be a given friction function and $w^\star \in \mathbb{R}^d$ be the unknown linear regressor. $n$ observations $(x_i, z_i)$ are produced as follows:*

1. *The features $x_i \in \mathbb{R}^d$ are arbitrary.*

2. *Each (unobserved) frictionless output satisfies $y_i = x_i^\top w^\star + \xi_i$ where $\xi_i \sim_{i.i.d.} \mathcal{N}(0, 1)$.*

3. *Observe the outputs with friction $(x_i, z_i = c(y_i))$.*

The goal is to use observations $(x_i, z_i), i \in [n]$ from Definition 8 and estimate $w^\star \in \mathbb{R}^d$ up to $L_2$-error $\varepsilon$ (the number of observations $n$ will depend on $1/\varepsilon$).

Similar to the case of coarse Gaussian mean estimation, some instances of linear regression with friction are information theoretically impossible to solve. Hence, we define the following notion of information preservation that quantifies identifiability.

**Assumption 4** (Information Preservation). *Let $\mathscr{P} := \left\{ c^{-1}(z) : z \in \mathbb{R} \right\}$ denote the partition induced by the friction function $c$. $\mathscr{P}$ is $\alpha$-information preserving with respect to $w^\star, x_1, \ldots, x_n$ for some $\alpha > 0$ if for any $w \in B(0, C)$,*

$$d_{\mathsf{TV}}\left(\mathcal{N}_{\mathscr{P}}\left(x_i^\top w, 1\right), \mathcal{N}_{\mathscr{P}}\left(x_i^\top w^\star, 1\right)\right) \geq \alpha \cdot d_{\mathsf{TV}}\left(\mathcal{N}\left(x_i^\top w, 1\right), \mathcal{N}\left(x_i^\top w^\star, 1\right)\right) .$$

In order to obtain computationally efficient algorithms, we focus on the case where the friction function induces a convex partition of $\mathbb{R}$. This is analogous to the case where we require the partition for coarse mean estimation to consist only of convex sets.

**Assumption 5** (Friction Function with Convex Pre-Image). *The friction function $c : \mathbb{R} \to \mathbb{R}$ has a convex (interval) pre-image $c^{-1}(z) \subseteq \mathbb{R}$ for any $z \in \mathbb{R}$ and we have access to $c$ via a pre-image oracle that outputs the boundaries $a \leq b \in \mathbb{R} \cup \{-\infty, +\infty\}$ of the interval pre-image $c^{-1}(z)$ given some $z \in \mathbb{R}$.*

We also impose the following standard assumptions for fixed-design linear regression, which appears even in the absence of friction.

**Assumption 6** (Bounded Parameters). *There are constants* $b, C, D > 0$ *such that* $\frac{1}{n} \sum_{i \in [n]} x_i x_i^\top \succeq b^2 I$, $\|w^\star\|_2 \leq C$, *and* $\|x_i\|_\infty \leq D$ *for all* $i \in [n]$.

For the sake of simplifying the proof, we further assume without loss of generality in Appendices E.1 to E.4 that $\|x_i\|_2 \leq D\sqrt{d} \leq 1$, by downscaling the covariates $x_i \leftarrow \frac{x_i}{D\sqrt{d}}$ and upscaling the input $w$ within the loss function $\ell_i(w) \leftarrow \ell_i(w \cdot D\sqrt{d})$. We undo this assumption in Appendix E.5.

With the assumptions in place, we now state the main result in this section.

**Theorem E.1.** *Let* $\varepsilon \in (0, 1/300)$. *Consider an instance of linear regression with friction and suppose Assumptions 4 to 6 holds with constants* $b, C, D > 0$. *There is an algorithm that outputs an estimate* $\widetilde{w}$ *satisfying* $\|\widetilde{w} - w^\star\|_2 \leq \varepsilon$ *with probability* $0.98$. *Moreover, the algorithm requires*

$$n = O\left( \frac{C^2 D^2 d}{\alpha^4 b^4 \varepsilon^2} \log^4\left( \frac{CDd}{\varepsilon} \right) \right)$$

*observations and terminates in* $\mathrm{poly}(n, T_s)$ *time, where* $T_s$ *is the time complexity of sampling from a 1-dimensional Gaussian distribution truncated to an interval (Assumption 7).*

We remark that Theorem E.1 recovers the required number of observations for ordinary least squares (OLS) without friction given a well-conditioned design matrix with smallest eigenvalue $\alpha^2 b^2$. Moreover, note that when the covariates are sampled from some well-behaved distribution, e.g., isotropic Gaussian, we have $D, b = \widetilde{\Theta}(1)$ with high probability, which recovers the tight $O(d/\varepsilon^2)$ sample complexity for Gaussian linear regression.

To simply our presentation, we use the following simplifying assumption.

**Assumption 7** (1-Dimensional Sampling Oracle). *There is an efficient sampling oracle that, given an interval* $[a, b] \subseteq \mathbb{R}$, *and a parameter* $\mu \in \mathbb{R}$ *describing a Gaussian distribution, outputs an unbiased sample* $y \sim \mathcal{N}(\mu, 1, [a, b])$.

We emphasize that Assumption 7 can be approximately implemented in polynomial time using Markov chains and defer the details to Appendix D.

**Technical Overview** Our main algorithm is a variant of gradient descent. This requires a few ingredients, which we collect in the following subsections.

- The iteration complexity of gradient descent depends on the second moment of our stochastic gradient oracle, which is difficult to analyze for general friction functions. In Appendix E.1, we define a feasible projection set containing $w^\star$ and consider the slightly restricted class of local friction functions.

- To ensure the iterates of gradient descent converge in Euclidean distance and not just in function value, we very that our likelihood function is convex and satisfies a local growth condition that is a strictly weaker form of strong convexity.

- Appendix E.3 describes a stochastic gradient oracle and develops a bound on its second moment for local friction functions.

- Appendix E.4 specifies the one-pass variant of gradient descent we employ for sums of functions and proves its guarantees.

- Appendix E.5 combines the ingredients collected throughout this section to derive an algorith for local frictions functions. Finally, the case of general friction functions is reduced to the local case.

### E.1 PROJECTION SET & LOCAL PARTITIONS

We take the projection set to be

$$K := \left\{ w \in \mathbb{R}^d : \|w\|_2 \leq C \right\} \tag{13}$$

where $C \geq 1$ is the known constant in Assumption 6. We can certainly efficiently project onto $K$ and this set contains $w^\star$ by assumption.

As in the previous sections, we analyze our algorithm under the special case of low-probability sets being singletons. Then we reduce the general case to this special case. We say that a friction function $c$ is *R-local* for some $R > 0$ if $|c^{-1}(z)| > 1$ implies that $c^{-1}(z) \subseteq [-R, R]$. That is, $c$ induces a partition of $\mathbb{R}$ that is $R$-local in the sense of Definition 5.

### E.2 CONVEXITY & LOCAL GROWTH

Given $n$ samples $(x_1, z_1), (x_2, z_2), \ldots, (x_n, z_n)$, for each $1 \leq i \leq n$, define

$$S_i = S(z_i) = c^{-1}(z_i).$$

Recall that the negative log-likelihood of $w$ given an observation $(x, z)$ is

$$\mathcal{L}(w; x, z) = -\log \mathcal{N}\big(S(z), x^\top w; 1\big).$$

Consequently, the log-likelihood of $w$ over all samples is

$$\mathcal{L}_{\mathscr{P}}(w) = \frac{1}{n} \sum_i \mathop{\mathbb{E}}_{S_i \sim \mathcal{N}_{\mathscr{P}}(x_i^\top w^\star, 1)} [-\log \mathcal{N}(S_i, \langle x_i, w \rangle ; 1)] = \frac{1}{n} \sum_i \ell_i(x_i^\top w).$$

Write $Q(i) := \mathcal{N}_{\mathscr{P}}\big(x_i^\top w^\star, 1\big)$. One can verify that its gradient and hessian are as follows.

$$\boldsymbol{\nabla} \mathcal{L}_{\mathscr{P}}(w) = \frac{1}{n} \sum_i \boldsymbol{\nabla} \ell_i(x_i^\top w) \cdot x_i = \frac{1}{n} \sum_i \mathop{\mathbb{E}}_{S_i \sim Q(i)} \left[ \langle x_i, w \rangle - \mathop{\mathbb{E}}_{u \sim \mathcal{N}(x_i^\top w, 1, S_i)} [u] \right] \cdot x_i,$$

$$\boldsymbol{\nabla}^2 \mathcal{L}_{\mathscr{P}}(w) = \frac{1}{n} \sum_i \boldsymbol{\nabla}^2 \ell_i(x_i^\top w) \cdot x_i x_i^\top = \frac{1}{n} \sum_i \mathop{\mathbb{E}}_{S_i \sim Q(i)} \left[ 1 - \mathop{\mathrm{Var}}_{u \sim \mathcal{N}(x_i^\top w, 1, S_i)} [u] \right] \cdot x_i x_i^\top.$$

These are sufficient to show that each $\ell_i$ and thus $\mathcal{L}(\cdot)$ is convex. Indeed, conditioning on a convex set $S_i \in \mathscr{P}(R)$ reduces the variance of the Gaussian distribution (Proposition B.2). It thus follows that $\mathrm{Var}_{u \sim \mathcal{N}(x_i^\top w, 1, S)}[u] \leq 1$ and hence, $\boldsymbol{\nabla}^2 \mathcal{L}(\cdot) \succeq 0$.

**Lemma E.2.** *Suppose Assumption 5 holds. Then each $\ell_i$ and thus $\mathcal{L}_{\mathscr{P}}(\cdot)$ is convex.*

Since $\mathcal{L}(\cdot)$ is convex, we can hope to use SGD to find a parameter $w$ that minimizes $\mathcal{L}(\cdot)$ in function value. Our next result shows that any $w$ maximizing $\mathcal{L}(\cdot)$ is also close to $w^\star$.

**Lemma E.3** (Quadratic Local Growth). *Suppose Assumptions 5 and 6 hold with constants $b, C > 0, D = \frac{1}{\sqrt{d}}$ and that $c(\cdot)$ is $\alpha$-information-preserving with respect to $w^\star$ and covariates $x_1, x_2, \ldots, x_n$ (Assumption 4). Then $\mathcal{L}$ satisfies an $(\alpha b/200, \alpha^2 b^2/600^2)$-local growth condition (Definition 6).*

*Proof.* From the definition of the log-likelihood, it can be verified that

$$\mathcal{L}(w) - \mathcal{L}(w^\star) = \frac{1}{n} \sum_i \mathsf{KL}(\mathcal{N}_{\mathscr{P}}(\langle x_i, w^\star \rangle, 1) \| \mathcal{N}_{\mathscr{P}}(\langle x_i, w \rangle, 1))$$

$$\geq \frac{2}{n} \sum_i d_{\mathsf{TV}}(\mathcal{N}_{\mathscr{P}}(\langle x_i, w^\star \rangle, 1), \mathcal{N}_{\mathscr{P}}(\langle x_i, w \rangle, 1))^2 \qquad \text{(Pinsker's inequality)}$$

$$\geq \frac{2\alpha^2}{n} \sum_i d_{\mathsf{TV}}(\mathcal{N}, (\langle x_i, w^\star \rangle, 1) \mathcal{N}(\langle x_i, w \rangle, 1))^2. \qquad \text{($\alpha$-information preserving)}$$

If $d_{\mathsf{TV}}(\mathcal{N}(\langle x_i, w^\star \rangle, 1), \mathcal{N}(\langle x_i, w \rangle, 1))$ is close to 0 for each $i$, then we can lower bound it in terms of the distance between the means $|\langle x_i, w^\star \rangle - \langle x_i, w \rangle|$ which, in turn, will enable us to get the required lower bound with respect to $\|w - w^\star\|_2$ using Assumption 6. However, we cannot guarantee that $d_{\mathsf{TV}}(\mathcal{N}(\langle x_i, w^\star \rangle, 1), \mathcal{N}(\langle x_i, w \rangle, 1))$ is close to 0 for all $i$. Instead, we show the following claim.

**Claim E.4.** *Suppose Assumption 6 holds with constant $b > 0$. Fix any $w$ satisfying $\mathscr{L}(w) - \mathscr{L}(w^\star) \leq \frac{\alpha^2 b^2}{600^2}$. There is a set $T \subseteq [n]$ of size at least $\left(1 - \frac{b^2}{2}\right) \cdot n$ such that for each $i \in T$,*

$$d_{\mathsf{TV}}(\mathcal{N}(\langle x_i, w^\star \rangle, 1), \mathcal{N}(\langle x_i, w \rangle, 1)) \leq \frac{1}{600} \, .$$

*Proof.* Let $T' \subseteq [n]$ be the set of all indices $i \in [n]$ such that

$$d_{\mathsf{TV}}(\mathcal{N}(\langle x_i, w^\star \rangle, 1), \mathcal{N}(\langle x_i, w \rangle, 1)) \geq \frac{1}{600} \, .$$

Toward a contradiction suppose that $|T'| \geq nb^2/2$. Our calculations above imply that

$$\mathscr{L}(w) - \mathscr{L}(w^\star) \geq \frac{2\alpha^2}{n} \cdot |T'| \cdot \frac{1}{600^2} \overset{(|T'| \geq nb^2/2)}{\geq} \frac{\alpha^2 b^2}{600^2} \, .$$

This is a contradiction and, hence, it must be the case that $|T'| \leq n/(2C)$. $\qquad\square$

Since $w$ satisfies that $\mathscr{L}(w) - \mathscr{L}(w^\star) < \frac{\alpha^2}{600C}$, Claim E.4 is applicable. Fix the set $T$ in Claim E.4. We have

$$\mathscr{L}(w) - \mathscr{L}(w^\star) \geq \frac{2\alpha^2}{n} \sum_{i \in T} d_{\mathsf{TV}}(\mathcal{N}(\langle x_i, w^\star \rangle, 1), \mathcal{N}(\langle x_i, w \rangle, 1))^2 \, .$$

Since for each $i \in T$, $d_{\mathsf{TV}}(\mathcal{N}(\langle x_i, w^\star \rangle, 1), \mathcal{N}(\langle x_i, w \rangle, 1)) \leq \frac{1}{600}$, Theorem 1.8 of Arbas et al. (2023) guarantees that every $i \in [T]$ satisfies

$$d_{\mathsf{TV}}(\mathcal{N}(\langle x_i, w^\star \rangle, 1), \mathcal{N}(\langle x_i, w \rangle, 1))^2 \geq \frac{\left(x_i^\top (w - w^\star)\right)^2}{200^2} \, .$$

Hence, whenever $\mathscr{L}(w) - \mathscr{L}(w^\star) \leq \alpha^2/600C$, we have

$$\mathscr{L}(w) - \mathscr{L}(w^\star) \geq \frac{2\alpha^2}{200^2 n} \sum_{i \in T} \left(x_i^\top (w - w^\star)\right)^2 = (w - w^\star)^\top \left(\frac{2\alpha^2}{200^2 n} \sum_{i \in T} x_i x_i^\top\right) (w - w^\star) \, .$$

Finally, the desired result follows because

$$\frac{1}{n} \sum_{i \in T} x_i x_i^\top = \frac{1}{n} \sum_{i \in [n]} x_i x_i^\top - \frac{1}{n} \sum_{i \notin T} x_i x_i^\top$$

$$\succeq \left(b^2 - \frac{n - |T|}{n}\right) \cdot I$$

$$\text{(as } \max_i \|x_i\|_2 \leq 1, \ x_i x_i^\top \preceq \|x_i\|_2^2 \cdot I, \text{ and } \textstyle\sum_{i \in [n]} x_i x_i^\top \succeq nb^2 \cdot I)$$

$$= \frac{b^2}{2} I \, . \qquad\qquad \text{(as } |T| = \left(1 - \frac{b^2}{2}\right) \cdot n)$$

$$\qquad\square$$

### E.3 BOUNDING THE SECOND MOMENT OF STOCHASTIC GRADIENTS

The objective function $\mathscr{L}$ is of the form

$$\frac{1}{n} \sum_{i \in [n]} \ell_i(x_i^\top w) = \frac{1}{n} \sum_{i \in [n]} \mathbb{E}_{S_i} \left[\mathscr{L}(x_i^\top w; S_i)\right] \, ,$$

where the gradient of $\ell_i(x_i^\top w) = \mathbb{E}_{S_i} \left[\mathscr{L}(x_i^\top w; S_i)\right]$ is of the form

$$\boldsymbol{\nabla}_w \ell_i(x_i^\top w) \cdot x_i = \left(\mathbb{E}_{S_i \sim \mathcal{Q}(i)} \left[\langle x_i, w \rangle - \mathbb{E}_{u \sim \mathcal{N}(x_i^\top w, 1, S_i)} [u]\right]\right) \cdot x_i \, .$$

Here $\mathcal{Q}(i) = \mathcal{N}_\mathscr{P}\left(x_i^\top w^\star, 1\right)$ where $\mathscr{P}$ is the partition of $\mathbb{R}$ induced by the friction function $c$.

**Lemma E.5.** *Suppose Assumptions 5 and 6 hold with constants $b, C > 0, D = \frac{1}{\sqrt{d}}$ and that $c$ is an $R$-local friction function. Given $w \in K$ in the projection set (Equation (13)) and a fixed covariate $x_i$, there is an algorithm that consumes a sample $S_i \sim \mathcal{N}_{\mathscr{P}}\left(x_i^\top w^\star, 1\right)$ and makes a single call to the sampling oracle (Assumption 7) to output a random variable $\widetilde{g}_i$ such that*

- *(i) $\widetilde{g}_i$ is an unbiased estimate of $\ell_i'(\cdot)$ with $\mathbb{E}[\widetilde{g}_i^2] = O(R^2 + C^2)$.*

- *(ii) $g_i := \widetilde{g}_i \cdot x_i$ is an unbiased estimate of $\boldsymbol{\nabla}_w \mathbb{E}_{S_i \sim \mathcal{Q}(i)}\left[\mathscr{L}(x_i^\top w; S_i)\right]$ with $\mathbb{E}\left[\|g_i\|_2^2\right] = O(R^2 + C^2)$.*

Before stating the proof, we note an application of Jensen's inequality yields as a corollary that $|\ell_i'|^2 = \left|\mathbb{E}\left[\widetilde{g}_i^2\right]\right|^2 = O(R^2 + C^2)$. In other words, $\ell_i$ is $L$-Lipschitz for $L = O(R + C)$.

*Proof.* Given $S_i \sim \mathcal{Q}(i)$, we can obtain a stochastic gradient for $\ell_i'$ by sampling $u \sim \mathcal{N}(x_i^\top w, 1, S_i)$, and outputting $\widetilde{g}_i = (x_i^\top w - u)$. Moreover, $g_i := \widetilde{g}_i \cdot x_i$ is a stochastic gradient for $\mathbb{E}_{S_i \sim \mathcal{Q}(i)}\left[\mathscr{L}(x_i^\top w; S_i)\right]$.

To bound the second moment of $\widetilde{g}$, we first note that

$$\widetilde{g}^2 \leq 2\left|x_i^\top w\right|^2 + 2|u|^2.$$

The first term can be deterministically bounded.

$$\left|x_i^\top w\right|^2 \leq \|w\|_2^2 \cdot \|x_i\|_2^2 \leq C^2. \qquad (\|x_i\|_2 \leq 1)$$

The second term can be bounded using $R$-locality as follows.

$$\left\|\underset{S_i \sim \mathcal{Q}(i)}{\mathbb{E}} \underset{u \sim \mathcal{N}\left(x_i^\top w, 1, S_i\right)}{\mathbb{E}}[u]\right\|^2$$

$$\leq \underset{S_i \sim \mathcal{Q}(i)}{\mathbb{E}} \underset{u \sim \mathcal{N}\left(x_i^\top w, 1, S_i\right)}{\mathbb{E}}\left[|u|^2\right] \qquad \text{(Jensen's inquality)}$$

$$= \underset{S_i \sim \mathcal{Q}(i)}{\mathbb{E}} \underset{u \sim \mathcal{N}\left(x_i^\top w, 1, S_i\right)}{\mathbb{E}}\left[|u|^2 \cdot \chi_{S_i \subseteq [-R,R]}\right] + \underset{S_i \sim \mathcal{Q}(i)}{\mathbb{E}} \underset{u \sim \mathcal{N}\left(x_i^\top w, 1, S_i\right)}{\mathbb{E}}\left[|u|^2 \cdot \chi_{S_i \cap [-R,R] = \varnothing}\right]$$

$$= \underset{S_i \sim \mathcal{Q}(i)}{\mathbb{E}} \underset{u \sim \mathcal{N}\left(x_i^\top w, 1, S_i\right)}{\mathbb{E}}\left[|u|^2 \cdot \chi_{S_i \subseteq [-R,R]}\right] + \underset{S_i \sim \mathcal{Q}(i)}{\mathbb{E}} \underset{u \sim \mathcal{N}\left(x_i^\top w^\star, 1, S_i\right)}{\mathbb{E}}\left[|u|^2 \cdot \chi_{S_i \cap [-R,R] = \varnothing}\right]$$

$$\qquad \qquad (R\text{-locality})$$

$$\leq R^2 + \underset{u \sim \mathcal{N}\left(x_i^\top w^\star, 1\right)}{\mathbb{E}}\left[|u|^2\right]$$

$$= R^2 + 1 + \left|x_i^\top w^\star\right|^2 \qquad \text{(second moment of Gaussian)}$$

$$\leq R^2 + 1 + C^2.$$

Hence, the second term can be bounded by $O(R^2 + C^2)$. Combining the two bounds yields a bound on $\mathbb{E}[\widetilde{g}^2]$.

The bound on the second moment of $g_i := \widetilde{g}_i \cdot x_i$ follows since $\|x_i\|_2 \leq 1$. $\qquad \square$

### E.4 One-Pass Projected Stochastic Gradient Descent for Functions satisfying Quadratic Growth

One subtlety is that we are attempting to minimize an average of functions but we can only use each sample once to compute gradients. This is because we only have access to a single coarse observation $S_i$ for each covariate $x_i$. Thus, we need a convergence analysis of PSGD under sampling without replacement, or, in other words, a one-pass algorithm.

**Theorem E.6** (Corollary 1 in Shamir (2016) combined with Theorem 3.1 in Hazan (2016)). *Let $F(w) = \frac{1}{n}\sum_{i\in[n]} F_i(w)$ be an average of convex functions $F_i : K \to \mathbb{R}$ over a closed convex domain $K \subseteq \mathbb{R}^d$. Suppose further that $F_i = f_i(x_i^\top w)$ and*

   *(i) $K \subseteq B(0, C)$*

   *(ii) $f_i$ is L-Lipschitz*

   *(iii) $\|x_i\|_2 \leq 1$ for each $i \in [n]$*

   *(iv) we have access to unbiased gradient oracles $g_i$ for $\ell_i$ such that $\mathbb{E}\left[\|g_i(w)\|_2^2\right] \leq G^2$*

*Then after $1 \leq T \leq n$ iterations, projected stochastic gradient descent without replacement with constant step size $\gamma$ over a random permutation of $F_i$'s outputs an average iterate $\overline{w}$ satisfying*

$$\mathbb{E}\left[F(\overline{w}) - F(w^\star)\right] \leq \frac{C^2}{\gamma T} + G^2\gamma + \frac{2(12 + \sqrt{2})LC}{\sqrt{n}} \, .$$

*Here, the randomness is taken over the random permutation as well as the stochasticity of gradient oracles.*

We take the functions $f_i : [-C, C] \to \mathbb{R}$ from Theorem E.6 to be

$$f_i(u) = \mathop{\mathbb{E}}_{S_i \sim \mathcal{N}_{\mathscr{P}}\left(x_i^\top w^\star, 1\right)} \left[-\log \mathcal{N}(S_i, u; 1)\right] \, .$$

For an $R$-local friction function, we can bound the Lipschitz constant of each function $f_i$ by $O(R + C)$ using a similar calculation as Lemma E.5.

One difference in our setting compared to Shamir (2016) is that they assume access to deterministic gradient oracles. However, it is not hard to see that the analysis extends to the case of stochastic gradient oracles.

Our first attempt would be to run the PSGD algorithm for sampling without replacement (Theorem E.6). For $R$-local partitions, we can bound the second moment of the gradient oracle as $G^2 = O(R^2 + C^2)$ (Lemma E.5). In order to recover $w^\star$ up to $\varepsilon$-Euclidean distance, we would like an $O(\alpha^2\varepsilon^2)$-optimal solution in function value (Claim E.4). Naïvely applying Theorem E.6 would yield an $\Omega(1/\varepsilon^4)$ gradient steps. Instead, we design an alternative one-pass iterative PSGD algorithm (Algorithm 1) which requires $\widetilde{O}(1/\varepsilon^2)$ steps instead. We note that the algorithm (Algorithm 1) and accompanying analysis (Theorem E.7) are similar to the PSGD analysis by Kalavasis et al. (2025), but we need additional ideas to argue that the algorithm converges in the one-pass setting, where we can only process each observation once.

**Theorem E.7.** *Let $F\colon K \subseteq \mathbb{R}^d \to \mathbb{R}$ be an average of functions $F(w) = \frac{1}{n}\sum_{i=1}^n f_i(x_i^\top w)$ where $\ell_i$ is convex. Suppose that*

   *(i) $K \subseteq B(0, C)$*

   *(ii) $f_i$ is G-Lipschitz*

   *(iii) $\|x_i\|_2 \leq 1$ for each $i \in [n]$*

   *(iv) we have access to unbiased gradient oracles $g_i(w)$ for $\ell_i(x_i^\top w)$ such that $\mathbb{E}\left[\left\|g_i(x_i^\top w)\right\|_2^2\right] \leq G^2$*

   *(v) $F$ satisfies a $(\eta, \varepsilon)$-local growth condition (Definition 6).*

   *(vi) We are given an initial $\varepsilon_0$-optimal solution $w^{(0)} \in K$.*

*There is an iterative PSGD algorithm (Algorithm 1) which proceeds in one pass over a random permutation of data and outputs a $2\varepsilon$-optimal solution with probability $0.98$ if $n = \Omega\left(\frac{G^2}{\eta^2\varepsilon}\log^4\left(\frac{\varepsilon_0}{\varepsilon}\right)\right)$.*

The proof of Theorem E.7 relies on the following result.

---

**Algorithm 1** One-Pass Iterative PSGD($K, w_0, \varepsilon_0, g, \eta, \varepsilon$)

---

1: **Input:** Projection access to feasible region $K$, initial point $w^{(0)} \in K$, $\varepsilon_0 \geq F(w^{(0)}) - F(w^\star)$, gradient oracles $g_i$ for $f_i(x_i^\top \cdot), i \in [n]$ with $\mathbb{E}\left[\|g_i\|_2^2\right] \leq G^2$, local growth rate $\eta > 0$, desired accuracy $\varepsilon > 0$
2: **Output:** $2\varepsilon$-optimal solution with probability 0.98
3: Set $\tau \leftarrow \lceil \log_2(\varepsilon_0/\varepsilon) \rceil$
4: Set $C_0 \leftarrow \frac{2\varepsilon_0}{\eta\sqrt{\varepsilon}}$
5: Set $\gamma_0 \leftarrow \frac{\varepsilon_0}{300G^2\tau}$
6: Set $T \leftarrow \frac{4 \cdot 300^2 G^2 \tau^2}{\eta^2 \varepsilon}$
7: $\sigma \leftarrow$ uniform random permutation of $[n]$
8: **for** $\ell = 1, \ldots, \tau$ **do**
9:     $\gamma_\ell \leftarrow 2^{-\ell}\gamma_0$
10:    $C_\ell \leftarrow 2^{-\ell}C_0$
11:    $w^{(\ell,0)} \leftarrow w^{(\ell-1)}$
12:    **for** $t = 1, \ldots, T$ **do**
13:        $j \leftarrow \sigma(\ell \cdot (T-1) + t)$         ▷ next index of gradient oracles in permutation
14:        $w^{(\ell,t)} \leftarrow \Pi_{K \cap B(w^{(\ell-1)}, C_k)}\left(w^{(\ell,t-1)} - \gamma_\ell \cdot g_j(w^{(\ell,t-1)})\right)$
15:    **end for**
16:    $w^{(\ell)} \leftarrow \frac{1}{T}\sum_{t=1}^{T} w^{(\ell,t)}$
17: **end for**

18: **Return** $w^{(\tau)}$

---

**Proposition E.8** (Lemma 1 in Xu et al. (2019)). *Suppose $F \colon \mathbb{R}^d \to \mathbb{R}$ satisfies a $(\eta, \rho)$-local growth condition. Then for any $w \in \mathbb{R}^d$,*

$$\|w - w_\rho^\dagger\|_2 \leq \frac{F(w) - F(w_\rho^\dagger)}{\eta\sqrt{\rho}} \, .$$

*Proof of Theorem E.7.* Define $\varepsilon_\ell \coloneqq 2^{-\ell}\varepsilon_0$. We will argue that $F(w^{(\ell)}) - F(w^\star) \leq \varepsilon_\ell + \varepsilon$ with probability $1 - 1/100\tau$ conditioned on $F(w^{(\ell-1)}) - F(w^\star) \leq \varepsilon_{\ell-1} + \varepsilon$. The claim follows then by a union bound over $\tau$ stages.

The base case for $\ell = 0$ holds by assumption. We have

$$C_\ell = \frac{\varepsilon_{\ell-1}}{\eta\sqrt{\varepsilon}} \qquad \text{and} \qquad \gamma_\ell = \frac{\varepsilon_\ell}{300G^2\tau} \, .$$

By Proposition E.8, we have

$$\left\|(w^{(\ell-1)})_\varepsilon^\dagger - w^{(\ell-1)}\right\|_2 \leq \frac{1}{\eta\sqrt{\varepsilon}}(F(w^{(\ell-1)}) - F(w^{(\ell-1)})_\varepsilon^\dagger) \leq \frac{\varepsilon_{\ell-1}}{\eta\sqrt{\varepsilon}} \leq C_\ell \, .$$

If $\ell = 1$, we can directly apply Theorem E.6 on the $\ell$-th stage of PSGD to deduce that

$$\begin{aligned}
\mathbb{E}[F(w^{(\ell)}) - F((w^{(\ell-1)})_\varepsilon^\dagger)] &\leq \frac{C_\ell^2}{\gamma_\ell T} + G^2\gamma_\ell + \frac{30GC_\ell}{\sqrt{n}} \\
&= \frac{\varepsilon_\ell}{300\tau} + \frac{\varepsilon_\ell}{300\tau} + \frac{\varepsilon_\ell}{300\tau} \\
&= \frac{\varepsilon_\ell}{100\tau} \, .
\end{aligned}$$

An application of Markov's inequality would then yield the desired result with probability $1 - \frac{1}{100\tau}$.

However, as our analysis conditions on each of the previous $\ell - 1$ stages, we cannot directly apply Theorem E.6 for $\ell > 1$ since the order of the permutation is not independent when conditioning on prior stages. Instead, we remark that the analysis of Theorem E.6 is equivalent to

1) randomly grouping the $n$ datapoints into $n/T$ subsets of size $T$ by repeating sampling $T$ points without replacement,

2) selecting one such subset uniformly at random, and

3) executing a one-pass PSGD on a random permutation of this selected group.

Thus, if $n \geq 100\tau^2 T$, then at the $\ell \leq \tau$ stage, we can perform our analysis conditioning on not selecting any of the at most $\tau$ subsets in the second step of the experiment above. Thus, the $\ell$-th stage still succeeds with probability $1 - \frac{2}{100\tau}$. $\qquad\square$

## E.5 Putting It Together

Applying Theorem E.7 to our setting with $(\eta, \rho) = (\alpha b / 200, \alpha^2 b^2 / 600^2)$ yields the following result. Note that compared to naïvely applying Theorem E.6, Theorem E.7 requires $O(1/\varepsilon^2)$ fewer observations.

**Proposition E.9.** *Let $\varepsilon \in (0, 1/300)$. Consider an instance of linear regression with friction and suppose Assumptions 4 to 6 holds with constants $b, C > 0, D = \frac{1}{\sqrt{d}}$. Suppose further that the partition $\mathscr{P}$ of $\mathbb{R}$ induced by the friction function is $R$-local. There is an algorithm that outputs an estimate $\widetilde{w}$ satisfying $\|\widetilde{w} - w^\star\|_2 \leq \varepsilon$ with probability $0.98$. Moreover, the algorithm requires*

$$n = O\left(\frac{R^2 + C^2}{\alpha^4 b^4 \varepsilon^2} \log^4\left(\frac{GC}{\varepsilon}\right)\right)$$

*observations and terminates in $\mathrm{poly}(n, T_s)$ time, where $T_s$ is the time complexity of sampling from a 1-dimensional Gaussian distribution truncated to a partition $P \in \mathscr{P}$ (Assumption 7).*

We now explain how to handle the more general condition that $\|x_i\|_\infty \leq D$ as well as remove the assumption of $R$-locality.

On a high level, We downscale the covariates $\overline{x}_i \leftarrow \frac{x_i}{D\sqrt{d}}$ while taking the scaled function $\overline{\ell}_i(x_i^\top w) := \ell_i(x_i^\top w \cdot D\sqrt{d})$. This does not affect information preservation or quadratic growth since $\overline{\ell}_i(\overline{x}_i^\top w) = \ell_i(x_i^\top w)$ are identical as functions of $w$. However, this increases the second moment of the gradient oracle by $\overline{G}^2 \leftarrow G^2 D^2 d$ as well as the Lipschitz constant of $\overline{\ell}_i$ by $\overline{L} \leftarrow GD\sqrt{d}$.

As for the $R$-locality assumption, we see that with probability $0.99$, the (unobserved) dependents without friction will not exceed $CD\sqrt{d} + O(\log n)$. Hence, we can essentially simulate an $R$-local partition for $R = CD\sqrt{d} + O(\log n)$.

We are now ready to prove the more general Theorem E.1. The proof details are similar to that of Theorem 3.2, where we approximately implement the algorithm for an $R$-local friction function using samples from a general friction function.

*Proof of Theorem E.1.* Suppose the friction function $f$ induces a general convex partition $\mathscr{P}$. Since $|x_i^\top w^\star| \leq CD\sqrt{d}$, choosing $R = CD\sqrt{d} + O(\log(n))$ means that any $n$-sample algorithm will not observe a sample $S_i \sim \mathcal{N}_\mathscr{P}(x_i^\top w^\star, 1)$ such that $S_i \cap [-R, R] = \varnothing$ with probability $0.99$ when the hidden constant in $R$ is sufficiently large. Define the partition $\mathscr{P}(R)$ of $\mathbb{R}$ given by

$$\mathscr{P}(R) := \{P \cap [-R, R] : P \in \mathscr{P}, P \cap [-R, R] \neq \varnothing\} \cup \{\{x\} : x \notin [-R, R]\}.$$

Since $\mathscr{P}(R)$ is a refinement of $\mathscr{P}$, it must also be $\alpha$-information preserving with respect to the covariates. Consider the set-valued function $H_R : \mathscr{P} \to \mathscr{P}(R)$ given by

$$P \mapsto \begin{cases} P \cap [-R, R], & P \cap [-R, R] \neq \varnothing, \\ \{x\} \text{ for an arbitrary } x \in P, & P \cap [-R, R] = \varnothing. \end{cases}$$

By the choice of $R$, with probability $0.99$, any $n$-sample algorithm cannot distinguish between observations $P_i \sim \mathcal{N}_{\mathscr{P}(R)}(x_i^\top w^\star, 1)$ and $H_R(S_i)$ for $S_i \sim \mathcal{N}_\mathscr{P}(x_i^\top w^\star, 1)$. Moreover, we can sample from $P \cap [-R, R]$ as per Assumption 7.

Thus, we can simply run the algorithm from Proposition E.9 with input observations $(x_i, H_R(S_i))$ given samples $(x_i, S_i)$ and conclude by applying Markov's inequality to the guarantees of Proposition E.9.  □

# F  SIMULATIONS ON VARIANCE REDUCTION

The proofs of both our primary estimation algorithms, *i.e.*, coarse mean estimation (Theorem 3.2) and the linear regression with friction (Theorem 3.3), systematically depend on a critical theoretical component: the uniform variance reduction exhibited by Gaussian variables upon conditioning on convex truncations (Proposition B.2). We complement our our theoretical analysis with empirical evidence of variance reduction.

Specifically, we empirically test the variance-reduction condition that are used in establishing our algorithm's guarantees. To study this, we draw i.i.d. samples from Beta, Gaussian, Laplace, and Quartic families in 1D with interval truncation sets. Then, we compare empirical variances before and after truncation. Across all instances, the variance of the truncated samples is consistently smaller, suggesting that our algorithm could, in principle, be extended to other distribution families as well, at least in a single dimension.

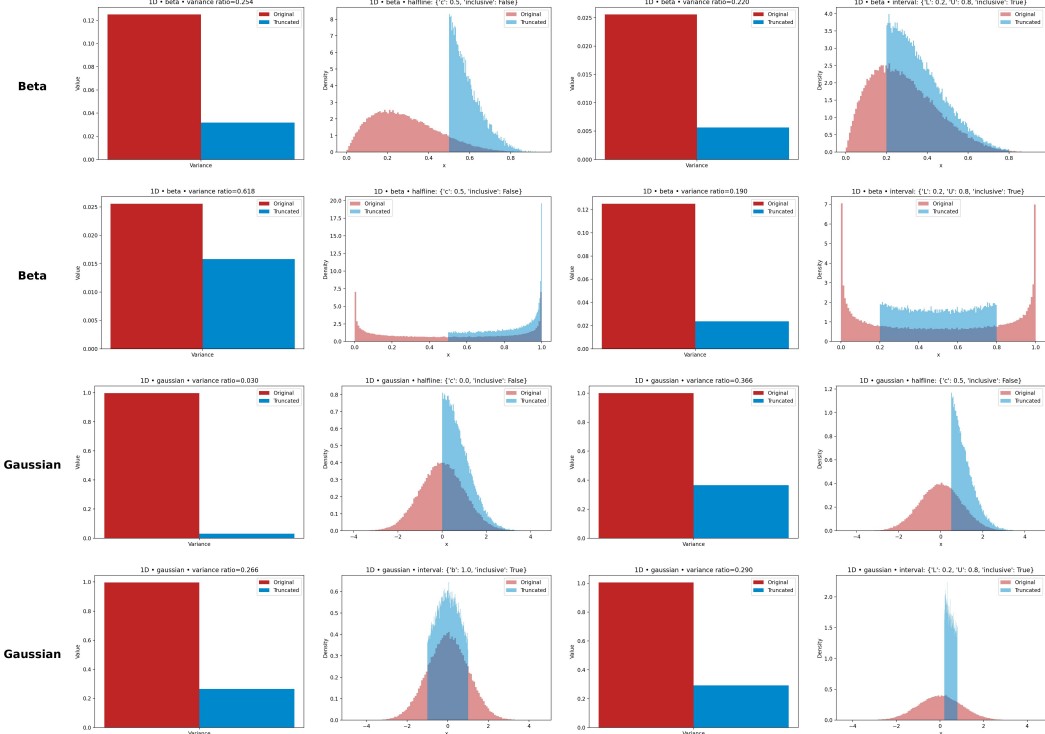

Figure 4: Each row shows a distribution (rows 1 and 2: Beta; rows 3 and 4: Gaussian). Columns (1,3): bar plots of empirical variances (red = original, blue = truncated) with variance ratio $r = \text{Var}_{\text{trunc}}/\text{Var}_{\text{orig}}$ annotated. Columns (2,4): histogram overlays (original vs. truncated) for half-line (col. 2) and interval $[L, U]$ truncations (col. 4). In all displayed runs, $r < 1$, indicating variance reduction under these convex truncations.

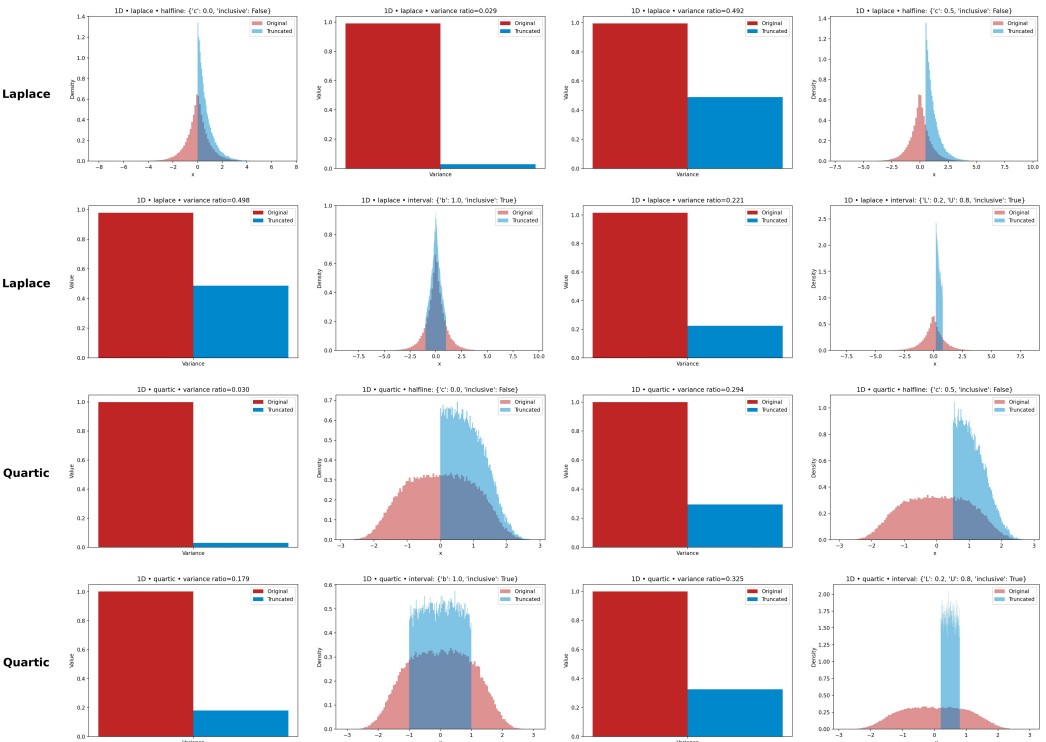

Figure 5: Same layout as Figure 4. Rows 1 and 2: Laplace; rows 3 and 4: Quartic (density $\propto e^{-(x-\mu)^4/s}$). Columns (1,3) report variance bars and $r$; columns (2,4) show the corresponding histogram overlays for half-line and interval $[L, U]$ truncations. The observed $r < 1$ across settings provides further evidence of practical variance reduction beyond the Gaussian case.

