# OpenReview forum: "Mean Estimation from Coarse Data: Characterizations and Efficient Algorithms"
_ICLR.cc/2026/Conference — ICLR 2026 Poster_

### Official Review · Reviewer_TRGK · 2025-10-30

**Soundness:** 2
**Presentation:** 3
**Contribution:** 3
**Rating:** 4
**Confidence:** 2

**Summary:**

This paper studies Gaussian mean estimation when observations are only available through coarse sets rather than exact samples.
It resolves two fundamental open questions from Fotakis et al. (2021) by (i) characterizing precisely when convex partitions are identifiable and (ii) providing the first polynomial-time algorithm for mean estimation under such identifiability.
The characterization shows that non-identifiability occurs if and only if the partition consists of parallel slabs in some direction.
The proposed algorithm performs stochastic gradient descent on a convex log-likelihood objective, with carefully controlled gradient moments and local strong convexity, achieving optimal $\tilde O(d/\omega^2)$ sample complexity.
The paper also illustrates the approach on linear regression with market friction, demonstrating both the conceptual generality and computational feasibility of learning from coarse data.

**Strengths:**

**Strengths:**

– The contributions are very clearly explained and well connected to prior work.

– The introduction is well written, motivating the problem both from statistical and practical perspectives.

– The problem studied is theoretically interesting, addressing a fundamental question at the intersection of statistics and learning theory.

– The characterization of identifiability (Theorem 3.1) is elegant and provides geometric intuition.

– The proposed algorithm is explicit and computationally efficient.

– The paper is well organized and easy to follow despite the technical depth.

**Weaknesses:**

**Weaknesses:  **

- The assumption to only consider Gaussian distributions is restrictive; it would strengthen the paper to discuss how the techniques might extend to broader exponential families, or to outline obstacles or required modifications.

- At several places the paper is lacking mathematical rigor. Most crucially, it is not clear what a  'probability distribution of a set' should be; please give a rigorous, measure-theoretic definition of \(\mathcal N_{\mathcal P}(\mu,I)\) as the pushforward of \(N(\mu,I)\) under the partition map and specify its sigma-algebra. As written, Definition 1 is hard to parse, and once \(\mathcal N_{\mathcal P}(\mu,I)\) is unclear, the meaning of KL divergence and TV distance between such objects is also opaque; moreover, the text sometimes treats distribution and  density as synonyms, which is mathematically incorrect.

- There seems to be a close relation to the literature on imprecise probability and set-valued observations (credal sets, lower and upper previsions), which is not discussed; adding a short paragraph contrasting your coarse partition with imprecise-probability frameworks would help position the work.

- The guarantees rely on the \(\varepsilon\)-information preservation, but the paper offers little guidance on verifying or estimating (\varepsilon\) in practice for a given partition beyond examples; a diagnostic or sufficient conditions (beyond non-slab or boundedness) would improve applicability.

**Questions:**

**Questions:**

- You write 'making estimation information-theoretically impossible.' What is the precise mathematical meaning of this statement? Please clarify in terms of identifiability or distinguishability of distributions.

- How can the definition of identifiability on page 2 be verified or practically tested for a given partition? Are there diagnostic criteria or examples that illustrate this?

- In Figure 1, it is not immediately clear why panel (a) represents a non-identifiable partition while (b) and (c) are identifiable. Could you provide more intuition or explanation for this already when the figure first appears? As it stands, the reader can only fully understand this after reading Theorem 3.1.

- Theorem 3.2 appears to be stronger than stated: you not only guarantee the existence of a polynomial-time algorithm, but in the proof you actually construct it explicitly. I suggest making this more prominent in the statement and the discussion.

---

> ### Author Response · Authors · 2025-11-22
>
> Thank you for your review. We answer your questions below.
>
> **R1 ``...extend to broader exponential families…’’** Yes, extending to exponential families is meaningful and we explicitly mention this as an interesting open problem in the conclusion along with a discussion outlining the key obstacles (Lines 477-481).
>
> **R2 ``...paper is lacking mathematical rigor…’’** We respectfully disagree. First, we note that the other reviewers found our exposition easy to follow. Second, we do provide specific definitions: we explicitly define the procedure that produces a coarse sample in Definition 1, from which measure-theoretic details can be deduced; we explicitly avoid measure-theoretic language to make the paper accessible to a broader computer science audience. For even greater clarity, we will add an additional clarifying remark about the coarse distribution as a push-forward measure.
>
> **R3 ``...relation to the literature on imprecise probability and set-valued observations…’’** Thank you for pointing out the potential connection to the literature on imprecise probability and set-valued observations (credal sets). While we have striven to contextualize our work broadly, we are not aware of any specific results in the mentioned literature that are directly relevant to our proposed framework or to coarse partitions. We are happy to include a discussion contrasting these frameworks in the final version, given appropriate pointers.
>
> **R4 On testing or estimating information preservation in practice.** This is an interesting question. However, we strongly believe that it is beyond the scope of this work. Indeed, it is common in, e.g., learning theory to first design algorithms relying on specific conditions (e.g., Gaussianity) and then other papers to design tests to empirically verify them.
>
> **R5 ``...What is the precise mathematical meaning of this statement?’’** We precisely define ``making estimation information-theoretically impossible’’ as (non-)identifiability in Definition 2.
>
> **R6 ``How can the definition of identifiability…be…tested for a given partition?...’’** Please see our response R4 above.
>
> ``R7 Could you provide more intuition…for [Figure 1]...? … the reader can only fully understand this after reading Theorem 3.1…’’* As we mention in this Figure’s caption, subfigure (a) is non-identifiable since ``its translation along the x-axis induces identical distributions of coarse samples.’’ Sub-figures (b) and (c) do not have this property but their identifiability is by no means obvious and requires a proof (as provided in Theorem 3.1). We will be happy to clarify this in the caption of the figure to avoid any confusion.
>
> **R8 Theorem 3.2 appears to be stronger than stated.** We do agree that the statement of Theorem 3.2 is slightly weaker than the result; however, this is a standard writing style in learning theory (and much of computer science). We will update to the more precise statement in the camera-ready version.

---

> > ### Comment · Reviewer_TRGK · 2025-11-26
> > **Response to rebuttal**
> >
> > I would like to thank the reviewers for their responses to my questions. I am willing to raise my score.

---

> > > ### Author Response · Authors · 2025-11-26
> > >
> > > We sincerely thank you for engaging with us and are very happy to see that the response led to an increase in your score.

---

### Official Review · Reviewer_wvZN · 2025-10-30

**Soundness:** 4
**Presentation:** 3
**Contribution:** 3
**Rating:** 8
**Confidence:** 4

**Summary:**

The paper considers the problem of mean estimation from coarse data. Concretely, the goal is to learn the location parameter, $\mu \in \mathbb{R}^d$, of a Normal distribution $\mathcal{N} (\mu, I)$ given access to i.i.d coarse samples from the distribution. Here, for a partitioning of $\mathbb{R}^d$, $\mathcal{P}$, one observes not a draw $X \thicksim \mathcal{N} (\mu, I)$ but rather the unique partition $P \in \mathcal{P}$ containing $X$. Prior work established that sample-efficient estimation is possible when $\mu$ is identifiable from coarse observations and the partitioning $\mathcal{P}$ consists only of convex sets. However, this prior work does not characterize \emph{when} $\mu$ is identifiable and if it is, when it can be estimated \emph{computationally} efficiently. The main contribution of this paper is to provide such a characterization.

To see why identification is not always possible, consider the special case where $d = 2$, and $\mathcal{P} = \{P_i\}_{i \in \mathbb{N}}$ with $P_i \coloneqq \{(x, y): i < y_i \leq i + 1\}$. It is clear that $\mu_1$ (the first coordinate) is not identifiable as any translation along the $x$ coordinate induces the same distribution over partitions. The main result of the paper establishes that these are the only scenarios where identification is not possible. That is, identification is impossible when the partition consists of a set of cylinders along a particular axis, with identification impossible along the axis. Furthermore, they show that a simple gradient descent-based algorithm on the log-likelihood of our observations enables computationally efficient estimation with a mild strengthening of this condition.

Technically, the main result follows from analyzing the (strict) convexity of the (log) likelihood function. When the partitions consist only of convex sets, the likelihood function can be easily shown to be convex. However, identification requires strict convexity. To do this, the starting point is an explicit characterization of the Hessian as the difference between $I$ and a weighted sum of conditional covariance matrices, which, from classical results in convex geometry, are dominated by $I$. To further extend these results to strict convexity, the authors carefully analyze when each of these conditional covariance matrices has variance $1$ along a particular direction. This, in turn, relies on more recent results in convex geometry.

Overall, this paper makes substantial progress on an interesting and well-motivated problem. In particular, the identifiability guarantees are natural and surprising. The proof, despite largely relying on classical techniques and results, is also quite non-trivial. I believe this paper would make an excellent contribution to the conference.

**Strengths:**

See main review

**Weaknesses:**

See main review

**Questions:**

See main review

---

> ### Author Response · Authors · 2025-11-22
>
> Thank you for your strong support. We are particularly happy that you found the problem interesting and thought that the results made substantial progress and were surprising!

---

### Official Review · Reviewer_THUX · 2025-11-01

**Soundness:** 3
**Presentation:** 4
**Contribution:** 3
**Rating:** 6
**Confidence:** 4

**Summary:**

This paper addresses the fundamental problem of Gaussian mean estimation when the observations are coarse data. The key theoretical contributions are resolving two fundamental questions related to identifiability and computational efficiency.
- Theorem 3.1: Intuitively, non-identifiability occurs if the partition is translation invariant in some direction. A convex partition $\mathcal{P}$ is not information preserving if and only if there is a unit vector $v \in \mathbb{R}^d$ such that almost every set $P \in \mathcal{P}$ is a slab in direction $v$.
- Theorem 3.2 provides a polynomial-time algorithm for finding $\epsilon$-accurate estimates of $\hat{\mu}$ for any convex, $\alpha$-information preserving partition $\mathcal{P}$.

The key assumption enabling the tractability and efficiency of Theorem 3.2 is that the partition $\mathcal{P}$ is convex and information-preserving.

**Strengths:**

The paper provides novel, complete solutions to the fundamental question in Gaussian mean estimation from coarse data. The proposed algorithm has optimal sample complexity.
- It establishes that a convex partition is non-identifiable if and only if almost every set is a slab in some common direction.
- It shows that identifiability is a property of the partition structure itself, as it does not depend on the true mean $\mu^{*}$.
- The algorithm achieves the optimal sample complexity of $\tilde{O}(d/\epsilon^2)$.
- $\mathcal{Z}(\mu)$ is proven to be convex because the individual component functions, $\mathcal{Z}_P(\mu) = -\log(\mathcal{N}(\mu, I; P))$, are convex for any convex set $P$ (Proposition B.2).
- The $\alpha$-information preservation assumption is used to prove that $\mathcal{Z}(\mu)$ satisfies a local growth condition around $\mu^{*}$ (Lemma B.3).
- This effectively means the function is locally strongly convex near the true mean $\mu^{*}$.
- This condition guarantees that finding an approximate minimizer $\hat{\mu}$ in function value ($\mathcal{Z}(\hat{\mu}) \approx$ $\mathcal{Z} (\mu^{*})$) ensures $\hat{\mu}$ is close to $\mu^{*}$ in $L_2$ distance.
- The natural gradient expression involves expectations over samples $x$ whose observed sets $P$ can be unbounded, leading to an unbounded variance in the gradient.
- They resolves this by introducing an idealized R-Local Partition $\mathcal{P}(R)$ (Definition 5).
- The final step is to remove the temporary R-Local Partition assumption and prove the theorem for a general convex partition $\mathcal{P}$.The argument sets the bounding radius $R$ large enough ($R = D + O(\log(md/\delta))$) so that the true, unobserved samples $x_i$ fall within $B_{\infty}(0, R)$ with high probability.
- The total sample complexity is derived by plugging the calculated local growth parameter ($\eta$) and the bounded second moment ($G^2$) into the PSGD convergence analysis (Theorem C.1)

**Weaknesses:**

- The polynomial time complexity relies on the existence of an efficient sampling oracle for a truncated Gaussian over the observed convex set $P$ (or $P \cap B_{\infty}(0, R)$). While this is achievable in $poly(d)$ time for polytopes using MCMC (Hit-and-Run) methods, these methods typically have high polynomial dependency and are usually slow in practice.
- The paper specifies the requirement for an efficient sampling oracle (Assumption 1) that outputs an unbiased sample $y \sim \mathcal{N}(\mu, I, P \cap B_{\infty}(0, R))$. This is an oracle for sampling from a truncated Gaussian distribution over a convex set. While the paper explicitly references the Hit-And-Run MCMC in Theorem D.1 as the general tool for implementing log-concave sampling over convex bodies (Appendix D), Langevin Monte Carlo (LMC), specifically Projected Langevin Monte Carlo (PLMC), is a highly relevant alternative method that could potentially be used. State-of-the-art PLMC methods often have better theoretical dependence on the dimension, such as $\tilde{O}(d)$ under certain conditions.
- The paper could be clearer on whether the entire partition $\mathcal{P}$ must have a manageable, unified representation, or just the observed set $P_i$. If the partition is dense (e.g., infinitely many sets), managing the required infrastructure for all $P \in \mathcal{P}$ could be impossible.

**Questions:**

See "weakness"

---

> ### Author Response · Authors · 2025-11-22
>
> We are happy to see that you liked our results and found them complete. We are responding to your specific comments and suggestions below, and hope that you would further strengthen your support for the paper.
>
> **”polynomial time complexity relies on the existence of an efficient sampling oracle”** That is correct, we do require an efficient sampling oracle (Assumption 1). This requirement of sampling oracle seems necessary to obtain stochastic gradients as the gradient of the log-likelihood is an expectation (which is NP-hard to compute exactly). Hence, if the sampling oracle runs in time $T_S$, then the running time of our algorithm is $O( d \varepsilon^{-2} (T_S+d) )$ (hiding dependence on $\alpha$ and $D$). It is indeed a very interesting open question to understand how small $T_S$ can be made for specific partitions, and as you mention, in some cases we can ensure $T_S=O(d)$, recovering a $O(d^2/\varepsilon^2)$ running time which is optimal even *without* any coarsening. We believe that this question of improving the sampling oracle is orthogonal to our work, which focuses on estimation provided a sampling oracle.
>
> **Langevin Monte Carlo (LMC)...could potentially be used.** This is an excellent point. We can indeed use LMC and other related methods to implement the gradient sampler as alternatives to Hit-and-Run MCMC; this leads to an improvement in the running time of our algorithm as we described above. We will add a discussion about this in the camera-ready version.
>
> **``...whether the entire partition P must have a manageable…representation…or just the observed set $P_i$...’’** We apologize for the confusion. We only require the observed sets $P_i$ to have a manageable representation. (This is because we only use a bound on the representation size to efficiently implement the gradient sampler; and we only need to sample gradients for the observed sets.)

---

> > ### Comment · Reviewer_THUX · 2025-11-24
> >
> > Thank you for the detailed response. I've read through your clarification regarding the sample oracle assumption, the potential use of LMC, and the representation required for the observed sets.
> > Everything is clear to me, I don't have any further questions at this time.
> >
> > I appreciate the effort you put into addressing each point, and I'm glad my comments were helpful.

---

> > > ### Author Response · Authors · 2025-11-24
> > >
> > > Thank you for the prompt follow-up!
> > >
> > >
> > > We are glad to hear that our response has fully addressed your concerns regarding the sampling oracle, the potential use of LMC, and the partition representation.
> > >
> > >
> > > As you know, even among positive reviews, the specific rating may influence the level of visibility at the conference (poster, spotlight, oral).  So, if you believe the merits of this work warrant it, we would be grateful if you could consider increasing your rating.
> > >
> > >
> > > Regardless, we appreciate your supportive comments in the review and reply.

---

> > > > ### Comment · Reviewer_THUX · 2025-11-25
> > > >
> > > > Increased contribution score from 3 to 4, and increased the score from 6 to 8.

---

> ### Author Response · Authors · 2025-11-26
>
> Thank you so much! We really appreciate your proactive responses!

---

### Official Review · Reviewer_RKSz · 2025-11-05

**Soundness:** 3
**Presentation:** 3
**Contribution:** 3
**Rating:** 8
**Confidence:** 3

**Summary:**

This paper studies the problem of estimating mean $\mu$ of a Gaussian distribution $\mathcal{N}(\mu, I)$ with identity covariance from coarse data, where the learner only observes which cell of a (potentially unknown) partition each sample falls into. The authors resolve two open questions from prior work (Fotakis et al., 2021):

 (i) Identifiability Characterization: For convex partitions, they provide a geometric characterization of when an instance is identifiable, i.e., the mean can be uniquely determined from observed samples. They show that an instance is non-identifiable if and only if almost all partition cells are parallel slabs in the same direction.

(ii) Efficient Algorithm:  They provide the first polynomial-time estimator for this problem under the assumptions that a bound $D$ on $\|| \mu \||$ is known and the partition is $\alpha$-information preserving (a stronger, quantitative version of identifiability assumption).

**Strengths:**

The coarse-data model is well motivated, and captures realistic phenomena like rounding, sensor quantization, and economic market friction.

The paper makes theoretical contributions by resolving two questions definitively: the identifiability characterization is clean and geometrically intuitive; while the estimator is both sample and computationally efficient, with time complexity polynomial in dimension $d$ and inverse desired accuracy $1/\epsilon$, and the bit/facet complexity of the descriptions of the observed partition cells (i.e., coarse samples).

The main text is well-written overall and easy to follow, and Section 4 provides a helpful overview of the intuition and core ideas behind the technical proofs.

The appendices appear thorough and the proof strategies combine tools from several areas creatively. For example, the authors use variance reduction and Prekopa-Leindler inequality for the identifiability characterization; as well as local partition reduction trick in bounding the gradient moments for the algorithmic result.

**Weaknesses:**

The estimator's sample complexity scales as $\widetilde{O}\left( \frac{d}{ \alpha^4 \epsilon^2} + \frac{dD^2}{\alpha^4} \right)$, where $D$ is known bound on $\|| \mu \||$. This is strictly worse than Fotakis et al.’s sample complexity of $\widetilde{O}\left(\frac{d }{\alpha^2 \epsilon^2}\right)$, which is independent of $D$. Also, I think claiming the sample complexity in the abstract as $\widetilde{O}\left(\frac{d }{\epsilon^2}\right)$ seems a little misleading, since this holds only in the constant regimes of $\alpha$ and $D$.



Minor: While each lemma and theorem is clear when read in isolation, I find the overall logical flow in the appendices (particularly from Appendix C onward) is quite dense and technically heavy. This is understandable given the paper’s theoretical depth and the secondary nature of these sections compared to Appendices A and B. But maybe adding a few sentences of contextual reminders or even a brief proof roadmap noting how the later technical lemmas connect back to the main results would help improve readability and navigation


Very minor (and less relevant since this is a theory-focused paper): The algorithm’s polynomial-time guarantee depends on access to an efficient sampling oracle. While the authors note in Appendix D that this can be implemented using MCMC schemes such as Hit-and-Run given a separation oracle, the practical runtime of these samplers is a concern. While this yields polynomial-time complexity, the high-degree polynomial dependencies (e.g., $d^{4.5}$) can be prohibitive for large or even moderate dimensions, limiting real-world applicability.

**Questions:**

If I understand correctly, the $\alpha^{-4}$ dependence in the sample complexity appears to stem from two $\alpha^{-2}$ factors: one from finding an $O(\alpha^2 \epsilon^2)$-optimal point in function value, and another from the $\Omega(\alpha^2)$ local strong convexity parameter. Do you think this dependence is fundamental for computationally efficient algorithms, or could it potentially be improved to $\alpha^{-2}$ (matching the sample complexity of Fotakis et al.) through a sharper analysis or an alternative scheme? A discussion on whether this reflects a intrinsic statistical–computational trade-off or an artifact of the current proof technique could be helpful.


I believe there is a minor typo in Remark 3. after Theorem 3.1. The sentence should likely be:
'The "almost every" quantifier in Theorem 3.1 is important: some convex partitions P contain non-slabs but are non-identifiable because the *non-slabs* have zero total volume.'

---

> ### Author Response · Authors · 2025-11-22
>
> Thank you for appreciating the results, exposition, and the proof strategy. We respond to your specific questions and comments below and hope you will retain your strong support for the paper.
>
> **On the estimator's sample complexity** Yes, our algorithm’s sample complexity is worse than that of Fotakis et al.: (1) We have an $\alpha^{-4}$ dependence instead of an $\alpha^{-2}$ dependence and (2) we require the diameter $D$ to be finite. We discuss these dependences further below. We note that the sample complexity in our work and in the work of Fotakis et al. matches in the terms of the dependence on $d$ and $\varepsilon$.
>
> **Dependence on $\alpha$:** In the submission, we did not try to optimize the dependence beyond ensuring that it is $\text{poly}(1/\alpha)$. We can improve the sample complexity to $(d/(\alpha^2\varepsilon^2) + dD^2/\alpha)$ at the cost of a polynomial blowup in the running time and present details below. We leave the question of matching the current running time while obtaining the optimal dependence on $\alpha$ as an interesting open question. (Note that this approach still requires the finiteness of $D$ to bound the second moment of the gradient and to get a finite warm-start.)
>
> > **Details about optimal sample complexity.** The approach is to use our analysis on the empirical log-likelihood instead of the population log-likelihood. Based on the result of Fotakis et al., we know that the minimizer of the empirical log-likelihood constructed using $O(d/(\varepsilon^2\alpha^2))$ samples is $\varepsilon$-close to $\mu^\star$. However, we do not have the guarantee that it is strongly convex. We can overcome this by using a variant of SGD for smooth convex functions. This, however, requires a polynomially larger number of iterations:  $O(1/(\varepsilon^4\alpha^4))$ instead of $O(1/(\varepsilon^2\alpha^2))$. This leads to an overall $\text{poly}(N)$ running time (however, the specific polynomial is worse than the polynomial in our main result). We would be happy to include a more detailed version of this argument in the camera-ready version.
>
> **Finiteness of $D$:** We require the finiteness of $D$ to obtain a warm-start for SGD and to bound the second moment of the stochastic gradients (which is required to bound the number of iterations of SGD). As Fotakis et al. have shown, this assumption can be removed if one does not require an efficient algorithm. Removing this assumption while having an efficient algorithm is indeed an interesting question.
>
> **”While each lemma and theorem is clear…overall logical flow…is…technically heavy.”** Thanks for the suggestion. As you noted, this is partly due to the nature of the results.  We will follow your suggestions and improve the exposition by using the narrative-based writing style as in Appendices A and B.
>
> **On sampling oracles.** Our stochastic gradient descent (SGD) algorithm relies on a sampling oracle to obtain stochastic gradients. If the sampling oracle runs in time $T_S$, then the running time of our algorithm is $O( d (\varepsilon^{-2}\alpha^{-4} + D^2\alpha^{-2}) (T_S+d) ).$ If $T_S=d^{4.5}$, this running time is indeed be a large polynomial (namely, $d^{5.5}$) and it is an interesting question to avoid the dependence on $T_S$. However, as Reviewer THUX01 mentions it is possible to improve the running time of sampling using Langevin Monte Carlo techniques and, for specific partitions, they can even have a running time of $T_S=O(d)$ leading to an overall running time of $O( d^2 (\varepsilon^{-2}\alpha^{-4} + D\alpha^{-1}) ).$ Provided $D/\alpha=O(1)$, this results in an $O(d^2/\varepsilon^2)$ running time which is optimal even without any coarsening.
>
> **”Typo in Remark 3 after Theorem 3.1”** Thanks, that is correct; we will fix it.

---

> > ### Comment · Reviewer_RKSz · 2025-11-24
> >
> > Thank you for the detailed response. I’ve read your clarification and I am happy to maintain my positive assessment.

---

### Meta-Review · Area_Chair_wb11 · 2026-01-08

**Summary:**

There is **strong overall consensus** among the reviewers that this paper makes a **substantial and high-quality theoretical contribution** to learning theory and high-dimensional statistics. Reviewers broadly agree that the paper resolves two fundamental open questions posed by Fotakis et al. (2021): it provides a geometric characterization of when a convex partition is identifiable, and it presents the first polynomial-time algorithm for computing $\varepsilon$-accurate estimates of the Gaussian mean from coarse samples generated by an unknown convex partition. The discussion during the rebuttal converged toward a clear acceptance, as multiple reviewers explicitly indicated score increases after the rebuttal.

**Reviewer Concerns:**

**Reviewer RKSz’s concerns:**

(1) The concern regarding the sample complexity of the proposed algorithm was addressed through additional clarification and discussion on its dependence on $\alpha$ and the finiteness of $D$. However, I agree with Reviewer RKSz that claiming the sample complexity in the abstract as $\widetilde{O}\left(\frac{d}{\varepsilon^2}\right)$ seems a little misleading, since the sample complexity of the proposed algorithm is in fact worse than that of Fotakis et al. (2021).

(2) The concern regarding the presentation was addressed by the authors’ commitment to improve the exposition in the revised version.

**Reviewer THUX’s concerns:**

(1) The concern regarding the efficient sample oracle assumption was addressed through more detailed explanations and discussion.

(2) The concern regarding the potential use of Langevin Monte Carlo (LMC) was addressed by including a discussion about this in the camera-ready version, as promised by the authors.

(3) The concern regarding the partition representation required for the observed sets was addressed through clarification.

**Reviewer wvZN’s concerns:**

None (no concerns were raised in the review).

**Reviewer TRGK’s concerns:**

(1) The concern regarding extension to broader exponential families was addressed, as discussed in the conclusion, along with an explanation of the key technical obstacles.

(2) The concerns regarding the mathematical rigor of Definition 1 and the precise mathematical meaning of a statement related to Definition 2 were addressed through additional clarification and discussion.

(3) The concern regarding related literature was addressed by including a discussion contrasting these frameworks in the final version, as promised by the authors.

(4) The concern regarding testing or estimating information preservation in practice was addressed by clarifying that this issue lies beyond the scope of the present work.

(5) The concern regarding Figure 1 was addressed through more detailed explanations and discussion.

(6) The concern regarding Theorem 3.2 was addressed by committing to a more precise statement in the camera-ready version.

**Reviewer Scores:**

**Reviewer RKSz** would likely maintain a score of 8, as the concerns raised in the review were adequately addressed in the rebuttal.

**Reviewer wvZN** would likely maintain a score of 8, as no concerns were raised in the review.

**Reviewer THUX** would increase the score from 6 to 8, as explicitly stated in the official comment, since the concerns raised were adequately addressed in the rebuttal.

**Reviewer TRGK** would raise the score (likely from 4 to 6), as explicitly stated in the response to the rebuttal, since the concerns raised were adequately addressed.

---

### Decision · Program_Chairs · 2026-01-26

Accept (Poster)